# A Comprehensive Genome-Wide Investigation of the Cytochrome 71 (*OsCYP71*) Gene Family: Revealing the Impact of Promoter and Gene Variants (Ser33Leu) of *OsCYP71P6* on Yield-Related Traits in Indica Rice (*Oryza sativa* L.)

**DOI:** 10.3390/plants12173035

**Published:** 2023-08-23

**Authors:** Bijayalaxmi Sahoo, Itishree Nayak, C. Parameswaran, Mahipal Singh Kesawat, Khirod Kumar Sahoo, H. N. Subudhi, Cayalvizhi Balasubramaniasai, S. R. Prabhukarthikeyan, Jawahar Lal Katara, Sushanta Kumar Dash, Sang-Min Chung, Manzer H. Siddiqui, Saud Alamri, Sanghamitra Samantaray

**Affiliations:** 1Crop Improvement Division, ICAR-National Rice Research Institute, Cuttack 753006, India; bijyalaxmisahoo7@gmail.com (B.S.); itishreenayak2616@gmail.com (I.N.); dr_hatanath_subudhi@yahoo.co.in (H.N.S.); cayalshiv@gmail.com (C.B.); jawaharbt@gmail.com (J.L.K.); skdash139@gmail.com (S.K.D.); smitraray@gmail.com (S.S.); 2Department of Botany, Ravenshaw University, Cuttack 753006, India; khirod555@gmail.com; 3Department of Botany, Utkal University, Bhubaneswar 751004, India; 4Department of Genetics and Plant Breeding, Faculty of Agriculture, Sri University, Cuttack 754006, India; 5Crop Protection Division, ICAR-National Rice Research Institute, Cuttack 753006, India; prabhukarthipat@gmail.com; 6Department of Life Science, Dongguk University-Seoul, Ilsandong-gu, Goyang-si 10326, Gyeonggi-do, Republic of Korea; smchung@dongguk.edu; 7Department of Botany and Microbiology, College of Science, King Saud University, Riyadh 11451, Saudi Arabia; mhsiddiqui@ksu.edu.sa (M.H.S.); saualamir@ksu.edu.sa (S.A.)

**Keywords:** allelic variants, cytochrome P450s, expression patterns, hormones, in/del markers, grain number, grain yield and plant development

## Abstract

The cytochrome P450 (CYP450) gene family plays a critical role in plant growth and developmental processes, nutrition, and detoxification of xenobiotics in plants. In the present research, a comprehensive set of 105 *OsCYP71* family genes was pinpointed within the genome of indica rice. These genes were categorized into twelve distinct subfamilies, where members within the same subgroup exhibited comparable gene structures and conserved motifs. In addition, 105 *OsCYP71* genes were distributed across 11 chromosomes, and 36 pairs of *OsCYP71* involved in gene duplication events. Within the promoter region of *OsCYP71*, there exists an extensive array of cis-elements that are associated with light responsiveness, hormonal regulation, and stress-related signaling. Further, transcriptome profiling revealed that a majority of the genes exhibited responsiveness to hormones and were activated across diverse tissues and developmental stages in rice. The *OsCYP71P6* gene is involved in insect resistance, senescence, and yield-related traits in rice. Hence, understanding the association between *OsCYP71P6* genetic variants and yield-related traits in rice varieties could provide novel insights for rice improvement. Through the utilization of linear regression models, a total of eight promoters were identified, and a specific gene variant (Ser33Leu) within *OsCYP71P6* was found to be linked to spikelet fertility. Additionally, different alleles of the *OsCYP71P6* gene identified through in/dels polymorphism in 131 rice varieties were validated for their allelic effects on yield-related traits. Furthermore, the single-plant yield, spikelet number, panicle length, panicle weight, and unfilled grain per panicle for the *OsCYP71P6-*1 promoter insertion variant were found to contribute 20.19%, 13.65%, 5.637%, 8.79%, and 36.86% more than the deletion variant, respectively. These findings establish a robust groundwork for delving deeper into the functions of *OsCYP71*-family genes across a range of biological processes. Moreover, these findings provide evidence that allelic variation in the promoter and amino acid substitution of Ser33Leu in the *OsCYP71P6* gene could potentially impact traits related to rice yield. Therefore, the identified promoter variants in the *OsCYP71P6* gene could be harnessed to amplify rice yields.

## 1. Introduction

The cytochrome P450 (CYP450) gene family plays a critical role in plant growth and developmental processes, nutrition, and detoxification of xenobiotics in plants [1,2,3,4,5,6,7]. In addition, CYP450 has been implicated in diverse metabolic reactions and targets a wide range of biological molecules. These biosynthetic reactions lead to diverse plant hormones, fatty acid conjugates, lignin, secondary metabolites, and numerous defensive compounds [8,9]. For instance, CYP706B1 and CYP82D113 participate in the biosynthesis of gossypol, while CYP71AV1 is implicated in the biosynthesis of artemisinin in plants [1,10]. The CYP450 genes have been identified and characterized in various organisms such as animals, bacteria, humans, and plants [1,2,3,4,5,11,12,13,14,15]. The plant CYP450 gene family is categorized into two primary groups: the A-type and the non-A type [16,17]. Further, several research groups reclassified them into eleven classes based on the available numerous genome sequences. The A-type is grouped into the CYP71 class, while the non-A type is subdivided into ten classes (CYP51, CYP72, CYP74, CYP85, CYP86, CYP97, CYP710, CYP711, CYP727, and CYP746) following a unified nomenclature system [17,18,19,20]. The members of CYP73, CYP84, and CYP98 gene families play a role in the phenylpropanoid biosynthetic pathway, contributing to the synthesis of various phenolic compounds. These compounds include substances such as flavonoids, suberin, polyphenols, and lignin [1,2,3,4,5]. Further, plant CYP450 enzymes have been linked to a wide array of metabolic pathways, giving rise to both primary and secondary metabolites [20]. For example, the majority of genes associated with the biosynthesis pathways of flavonoids and isoflavonoids are members of the CYP450 gene family. Notable examples include CYP73, CYP75A, and CYP93C [21]. Remarkably, the diversity in the P450 gene family had a significant impact on the emergence of novel metabolic pathways during the evolution of land plants. A classic example is cytochrome P450 93C, which is identified in leguminous crops and enzymes belonging to the CYP93C subfamily and participates in the legume-specific isoflavonoid biosynthesis pathway [22,23,24,25,26].

In rice, the gene *OsCYP71P6* (BGIOSGA018523) is a homolog of *CYP71A1* (LOC_Os12g16720). This gene encodes a CYP450 monooxygenase in rice that exhibits tryptamine 5-hydroxylase activity. Its function involves catalyzing the conversion of tryptamine into serotonin [27,28]. Phytoserotonin (5-hydroxytrptamine) is found in different tissues of plants, is involved in growth and development, and regulates tolerance to different biotic and abiotic stresses [27]. Further, the synthesis of serotonin through the amino acid tryptophan is highly conserved and distinct within plants. Consequently, alterations in the serotonin biosynthesis pathway have a direct impact on various plant developmental processes. A loss-of-function mutation of *OsCYP71P6*, involved in the activity of the tryptamine-5-hydroxylase gene and serotonin synthesis, resulted in brown plant hopper (BPH) and stem borer resistance in rice by altering pest feeding behavior. However, yield traits were also affected by the loss-of-function mutation of *OsCYP71P6* in rice [28]. Rice holds a prominent position as a staple food in countries across South Asia [29,30]. The popularity of rice varieties among farmers is greatly determined by their panicle architecture, including primary and secondary branches, the count of spikelets per panicle, stable yield, and milling properties [31]. In rice, several genes have been identified and characterized, such as *Gn1, DEP1, GS3, SPL18*, and *IPA1,* which regulate panicle architecture, grain number, and yield [32]. Additionally, many of the genes associated with yield also play a role in regulating other crucial traits in rice. For example, elevated expression of the IPA1 gene, which encodes OsSPL14 in young panicles, correlates with an increased number of primary branches [33] and blast disease resistance [34]. Similarly, the *Dense and Erect Panicle 1* (*DEP1*) gene regulates nitrogen use efficiency and yield [35]. Although the *OsCYP71P6* gene provides tolerance to insect pests, reduced serotonin differentially regulates senescence and negatively affects rice yield [28]. Gene sequence polymorphisms are common occurrences in both coding and promoter regions, leading to variations in gene transcription. These variations play a role in phenotypic adaptation among diverse rice varieties [36,37]. The availability of the rice 3K panel significantly aids in detecting single nucleotide polymorphisms and insertion–deletion (in/dels) within chromosomal regions or genes [38]. Previously, an insertion present in the 3′UTR of the TPP7 (*trehalose phosphate phosphatase 7*) gene was found to be associated with enhanced anaerobic response in rice [39].

Following recent advancements in DNA-sequencing technology, there has been a surge in the number of sequenced plant genomes. This has facilitated genome-wide analysis and characterization of the CYPP450 gene family in various crop species, including *Arabidopsis thaliana* [17,40], *Glycin max* [7], *Morus notabilis* [10], *Nicotiana tabacum* [41], and *Linum usitatissimum* [42]. The CYP450 gene family is extensive, and only a limited number of its members have undergone comprehensive structural and functional characterization. Additionally, there remains a scarcity of information regarding the CYP71 gene family in rice. The evolutionary lineage, gene and motif arrangement, cis-regulatory element involvement, and transcription kinetics of the CYP71 gene family in rice are currently unexplored. A comprehensive genome-wide investigation and analysis spanning various plants from different evolutionary branches is imperative. Such information could potentially enrich our comprehension of the evolutionary lineage and biological roles of the CYP71 gene family in the plant kingdom. The complete genome sequence of indica rice (*Oryza sativa* Indica) is publicly accessible, enabling us to conduct a comprehensive genome-wide analysis of the CYP71 gene family in rice [43]. Rice holds significance as both a vital cereal crop and a model plant in scientific research [29,30,44]. Hence, in this work, we conducted a comprehensive genome-wide analysis of the CYP71 gene family in rice, employing a range of computational tools. Further, we investigated the physical and biochemical characteristics, gene structure, motif composition, chromosomal distribution, gene duplication events, 3D structural aspects, and expression patterns of members of the *OsCYP71* gene family across different tissues and developmental stages. Our hypothesis centers on the notion that the insertion and deletion (in/dels) variations within the *OsCYP71P6* gene in rice could provide insights into the gene’s impact on yield-related traits. Our objective is to pinpoint specific in/dels variations unique to *OsCYP71P6* and validate these sequence polymorphisms to ascertain their effects on yield-related traits in rice. Furthermore, through linear regression models, we identified eight promoters and a gene variant (Ser33Leu) of *OsCYP71P6* that exhibited associations with spikelet fertility. Additionally, we validated various alleles of *OsCYP71P6*, identified via in/dels polymorphism in 131 rice varieties, to determine their effects on yield-related traits. The identified promoter and gene variants within *OsCYP71P6* hold potential for enhancing rice yield. Collectively, this study offers a valuable resource for further comprehending the specific functions of *OsCYP71* gene family members in rice.

## 2. Results

### 2.1. Genome-Wide Identification and Phylogenetic Tree Analysis of OsCYP Genes in Rice

In this study, a total of 105 CYP71 genes were successfully identified in rice (*Oryza sativa* Indica). These 105 *OsCYP71* genes were found to encode proteins ranging in length from 321 to 1165 amino acids long (Table 1 and Appendix A).

The molecular weight of the encoded protein is between 36.35 kda (OsCYP71K2) and 130.4 kda (OsCYP71AB12). The isoelectric point sizes ranged from 5.45 (OsCYP71K2) to 9.90 (OsCYP71AB13); among them, 82 proteins encoding OsCYP71 were identified by PI > 7, suggesting that most OsCYP proteins have a high proportion of basic amino acid. We observed GRAVY values for 105 OsCYP71 proteins between −0.409 (OsCYP71E2) and 0.197 (OsCYP71U3). Further, subcellular localization projections indicated that 100 OsCYP71 proteins were localized in the endomembrane system. The rest are mapped to plasma membrane (OsCYP71C1, OsCYP71Z4, OsCYP71V1, OsCYP71E2, and OsCYP71C3) and mitochondrial membrane (OsCYP71P6). In addition, we also examined the molecular weight of OsCYP71 to the pI to determine the distribution of OsCYP71 proteins (Appendix A). The outcomes of this analysis unveiled a pattern where the majority of OsCYP71 proteins share a comparable molecular weight and isoelectric point. To evaluate the evolutionary connections among *OsCYP71* genes, we employed CYP71 genes from Arabidopsis and tomato. Using MEGA 11 software, we constructed a phylogenetic tree that encompassed 105 CYP71 proteins from rice along with 42 CYP71 proteins from Arabidopsis and 43 CYP71 proteins from tomato (Figure 1, Appendix A).

The results showed that 105 *OsCYP71* genes could be divided into 12 subfamilies. Group VII had 74 members, whereas Groups I, II, III, IV, V, VI, VIII, IX, X, XI, and XII had 0, 8, 5, 0, 13, 2, 3, 0, 0, 0, and 0 members, respectively (Appendix A).

### 2.2. Chromosome Distribution and Gene Duplication Analysis of OsCYP71 Genes

To further investigate gene duplication events in rice, we performed a chromosomal mapping analysis of identified *OsCYP71* family genes. A total of 105 *OsCYP71* genes were mapped onto 11 chromosomes (Figure 2A). Chromosome 2 had the most *OsCYP71* genes (20 genes) and chromosome 7 had the least *OsCYP71* genes (1 gene). The 105 *OsCYP71* genes were distributed unevenly on 12 chromosomes of rice (Figure 2B).

The four *OsCYP71* genes were distributed across a genomic scaffold (Table 1). Gene duplication plays a crucial role in the evolution of gene families. In order to study the evolutionary relationship of *OsCYP71* genes, the gene duplication events of *OsCYP71* genes were analyzed in this study (Figure 3 and Appendix A).

A total of 36 duplicated gene pairs were detected (Appendix A). Further, the Ka/Ks score for these genes was less than one, indicating that there had been a strong purifying selection with slight changes in subsequent gene duplication. These findings highlight the fact that the *OsCYP71* family genes have undergone a conserved evolutionary process.

### 2.3. Gene Structure and Basic Motif Analysis of OsCYP71 Gene

Analyzing gene structure provides valuable insights into the conserved features and evolutionary distinctions of CYP proteins in rice. Through this analysis, we observed that the number of introns varied across different subfamilies (Figure 4).

The number of introns of all CYP subfamily genes ranged from zero to four, whereas most of the genes have between one and two introns. The maximum four introns were detected in *OsCYP71E2*, while *OsCYP71AD*, *OsCYP71C3*, *OsCYP71C4*, *OsCYP71AB7*, *OsCYP71W5*, *OsCYP71U3*, *OsCYP71T4*, *OsCYP71T7*, *OsCYP71T5*, *OsCYP71T8*, *OsCYP71T6*, and *OsCYP71T9* were intronless (Figure 4). Further, to investigate the characteristics of OsCYP71 proteins, we identified ten conserved motifs in OsCYP71 proteins (Figure 5A,B).

In this study, it was observed that motifs 1, 2, 3, 6, 7, and 10 exhibited a high degree of conservation across the majority of OsCYP71 proteins. Additionally, through amino acid sequence alignment, it was noted that OsCYP71 proteins contain conserved SRS and heme-binding motifs (Appendix A). We also conducted predictions for the 3D structure of OsCYP71 proteins (Appendix A). Numerous conserved motifs identified within these OsCYP71 family proteins are likely involved in the regulation of diverse metabolic reactions.

### 2.4. Promoter Element and GO Enrichment Analysis

To gain insights into the potential functions of the *OsCYP71* family genes, we employed the Plant-CARE online web server to analyze the 1500 bp upstream sequences of these genes. This analysis revealed the presence of numerous cis-regulatory elements, encompassing phytohormone responsiveness, light response, cell cycle regulation, circadian rhythm, seed-specific regulation, and responses to defense and stress (Figure 6A,B).

In terms of hormonal response, there are abscisic acid-responsive elements (ABRE), auxin-responsive elements (TGA-element, AuxRE, TGA-box), gibberellin-responsive elements (Tatc-box, GARE-motif, P-box, TATC-box), salicylic acid-responsive elements (TCA-element, SARE), methyl jasmonate-responsive elements (CGTCA-motif, TGACG-motif), and so on (Appendix A). MeJARE, ABRE, and AuxRE were found in the promoter regions of most genes, with MeJARE and ABRE being found in almost all *OsCYP71* genes. SARE- and GARE-motif elements were found in a few promoter regions of *OsCYP71* genes. Stress response includes elements related to anaerobic induction (ARE), low temperature response (LTR), drought induction (MBS), defense and stress responses (TC-rich repeats), and low temperature and salt stress (DRE). Among them, the ARE element was found in the promoter regions of 40 genes, the LTR element was found in the promoter regions of 38 genes with 1–2 elements, and TC-rich repeat elements were found in 21 gene promoter regions with 1–2 elements. In addition, MBS elements, considered drought induction-related elements, were also present in the promoter regions of 43 genes (Appendix A). These results suggest that the *OsCYP71* family genes may play an important role in plant development and stress responses through the regulation of multiple cis-regulatory elements in rice.

In order to further understand the function of *OsCYP71* family genes, we conducted a gene ontology (GO) enrichment analysis. Utilizing AgriGO, we effectively annotated and attributed GO terms to all OsCYP71 family genes. This annotation was subsequently validated using eggNOG-Mapper (Appendix A; Appendix A), which produced almost the same results as AgriGO. OsCYP genes showed enrichment in metabolic processes (GO:0008152) and oxidation-reduction processes (GO:0055114) in the biological process category (Appendix A), while in the molecular category, OsCYPs showed enrichment in catalytic activity (GO:0003824), binding (GO:0005488), electron carrier activity (GO:0009055), oxidoreductase activity (GO:0016491), monooxygenase activity (GO:0004497), tetrapyrrole binding (GO:0046906), and heme binding (GO:0020037) (Appendix A). Interestingly, no GO enrichments were identified within the cellular category. As a result, these outcomes strongly suggest that the OsCYP71 family genes are pivotal in various metabolic processes in rice.

### 2.5. Transcriptome Profiling of OsCYP71 Family Genes in Different Tissues and Phytohormone Treatments in Rice

To gain insights into the functions of *OsCYP71* genes, we examined their expression patterns across various tissues, developmental stages, and under hormone treatments. We sourced *OsCYP71* gene family expression data from the Rice Expression Profile Database (RiceXPro), which we then employed to construct heatmaps. Remarkably, among the 105 *OsCYP71* family genes, we identified differential expression across different tissues, developmental stages, and in response to hormone treatments (Figure 7A–C).

For instance, *CYP71X4* and *CYP71X12* expression was elevated in ovary, embryo, and endosperm tissue, while the expression of *CYP71X13P* was induced in lemma, palea, and endosperm tissue. Further, the expression of *CYP71X7* was highly induced in vegetative, reproductive, ripening leaf blade, vegetative, reproductive leaf sheath, ovary, embryo, and endosperm tissue, whereas the expression of *CYP71Z6*, *CYP71R2P*, *CYP71R1*, and *CYP71K12* was highly up-regulated in vegetative, reproductive, ripening leaf, vegetative, and reproductive leaf sheath tissue. *CYP71C19P* and *CYP71Z2* expression were increased in vegetative, reproductive, ripening leaf, vegetative, reproductive leaf sheath, root vegetative, and root reproduction tissue. Additionally, we also examined the expression pattern of *OsCYP71* in the shoots and roots under different plant hormone treatments such as auxin, gibberellin (GA), abscisic acid (ABA), cytokinin, jasmonic acid (JA), and brassionsteroid (BRS) (Figure 7B,C). *CYP71Q1* expression was elevated in the shoot in the ABA and JA treatments after 3 h, 6 h, and 12 h, while the expression level of *CYP71Z4* was induced in the shoot after 3 h, 6 h, and 12 h of GA treatment. The transcript levels of *CYP71C16*, *CYP71Z6*, *CYP71C19P*, *CYP71V2*, *CYP71X10*, *CYP71W1*, and *CYP71T1* were increased in the shoot after 3 h, 6 h and 12 h of JA treatment. Further, expression of *CYP71X10* increased in the shoot after 3 h, 6 h, and 12 h of BRS treatment, while *CYP71X11* was elevated in the shoot after 1 h, 3 h, 6 h, and 12 h of auxin treatment (Figure 7B). The expression of *CYP71X12* increased in the root after 1 h, 3 h, and 6 h of cytokinin treatment, whereas *CYP71X14*, *CYP71P1*, and *CYP71C16* were significantly induced in the root after JA treatment for 30 min, 1 h, and 3 h. Furthermore, *CYP71Z4*, *CYP71W1*, *CYP71K5*, *CYP71Z2*, *CYP71C20*, and *CYP71Z3* were highly up-regulated in the root after JA treatment for 3 h and 6 h (Figure 7C). Collectively, these results have shown that *OsCYP71* gene family members may participate in different developmental processes and responses to phytohormones in rice.

### 2.6. Gene Diversity Analysis of OsCYP71P6 Alleles

To examine gene variation, four in/del primers were validated in the one hundred and thirty-one different rice varieties. Out of four primers of the *OsCYP71P6* gene, two primers were polymorphic (*OsCYP71P6*-1; *OsCYP71P6*-4) and the other two were found to be monomorphic in rice varieties. The primer *OsCYP71P6-1* has two different allelic variants, having amplicon lengths of 320 bp and 350 bp (Appendix A). Among the 131 rice varieties examined, 113 displayed an amplicon length of 320 bp, while 18 varieties exhibited an amplicon length of 350 bp (Appendix A). Similarly, the *OsCYP71P6-4* gene also showed two different alleles, with 107 rice varieties displaying amplicon lengths of 380 bp and 24 rice varieties displaying lengths of 400 bp. Further, the in/del *OsCYP71P6-1* major allelic frequency was 0.8625, which was relatively higher than that of in/del *OsCYP71P6-4* (0.8167); overall, the mean major allelic frequency for both the markers was 0.8396 (Table 2).

Further, mean gene diversity for both the in/dels was found to be 0.2681, while the gene diversity of *OsCYP71P6-4* was 0.2992 and that of *OsCYP71P6-1* was 0.2370. Additionally, the PIC values of the *OsCYP71P6-1* and *OsCYP71P6-4* markers were 0.2090 and 0.2545, respectively, with a mean PIC value of 0.2317 (Table 2).

### 2.7. Grouping of Rice Varieties Based on OsCYP71P6 Alleles

The unweighted pair group method with arithmetic averaging (UPGMA) was employed for distance-based diversity analysis. This analysis revealed the presence of two primary clusters, designated as Cluster A and Cluster B (Figure 8).

Furthermore, Cluster A is subdivided into two branches, namely, A1 and A2, which are further fragmented into several minor clusters. Similarly, Cluster B was also subdivided into two sub clusters (Figure 8). Cluster A encompasses a total of 24 distinct rice varieties, while Cluster B is the largest cluster, comprising 107 different rice varieties. The model-based structure analysis revealed genetic relationships between different rice varieties based on the *OsCYP71P6* alleles. The optimal K value was found to be 2 (K = 2), indicating that the 131 rice varieties were grouped into two subpopulations (Figure 9A–C).

Additionally, within subpopulation I, a total of 29 rice varieties were identified, while subpopulation II encompassed 60 rice varieties. Moreover, 42 rice varieties were classified under the admixture category in the analysis. Notably, the analysis employed a threshold value of 60% (Appendix A). AMOVA (analysis of molecular variance), based on F-statistics, revealed that 81% of the genetic variation existed between populations, while the remaining 19% of variation was observed within the population (Table 3).

### 2.8. Descriptive Statistics of Yield-Related Traits for Different Alleles of OsCYP71P6

The descriptive statistics for the seven different traits studied in the one hundred and thirty-one rice varieties are shown in Table 4.

The varieties were grouped based on the alleles of the two polymorphic in/dels of *OsCYP71P6*. The observed trait values spanned ranges for spikelet number (163.99 to 189.93), single-plant yield (30.71 to 38.48 g), panicle weight (3.29 to 3.61 g), number of productive tillers (10.29 to 11.98), panicle length (25.89 to 27.36 cm), unfilled grain (29.2 to 47.65), filled grain (133.91 to 142.61), and 100 seed weight (2.27 to 2.4 g). The kurtosis values for these traits ranged from −0.83 to 7.58. Notably, among the studied traits, unfilled grain exhibited the highest kurtosis value (7.58). Similarly, the skewness of various traits ranged from −0.78 to 1.85.

### 2.9. OsCYP71P6 Allelic Difference in Phenotypic Traits

Further, we examined the allelic variation in the *OsCYP71P6* gene and associations with phenotypic traits. Among the two *OsCYP71P6* primers, the alternate alleles of *OsCYP71P6-4* showed statistically significant difference for traits, namely, panicle length, single-plant yield, panicle weight, number of spikelets, and unfilled grain with significant *p* values of 0.002, 0.005, 0.05, 0.004, and 0.001, respectively (Table 5).

In terms of percent difference in single-plant yield, number of spikelets, panicle length, panicle weight, and unfilled grain for *OsCYP71P6-4*, the 400 bp allele in/del primer was found to contribute 20.19%, 13.65%, 5.37%, 8.79%, and 36.86% more, respectively, than the 380 bp allele in the studied rice varieties. The average single-plant yield was found to be 30.71 ± 11.38 g and 38.48 ± 13.87 g for alleles of 380 bp and 400 bp lengths, respectively, with a *p* value of 0.005. Similarly, the mean number of spikelets was found to be 163.99 ± 54.16 and 189.93 ± 43.86 for 380 bp and 400 bp allelles, respectively, and difference was found to be significant (*p* value: 0.004). The panicle weight trait also showed statistically significant differences between the alleles (*p* value: 0.05), with the mean values of 3.29 ± 0.95 and 3.61 ± 0.84 for 380 bp and 400 bp alleles, respectively. Likewise, mean panicle length between the alleles was 25.89 ± 2.81 and 27.36 ± 2.26 for 380 bp and 400 bp alleles, respectively, with a *p* value of 0.002. The 380 bp and 400 bp alleles showed a mean unfilled grain value of 30.08 ± 15.86 and 47.65 ± 28.66, respectively, and a significant *p* value (0.001). In contrast, the 350 bp allele of the *OsCYP71P6-1* primer contributes 17.64% and 14.07% in terms of single-plant yield and number of productive tillers, respectively, relative to the 320 bp allele. Furthermore, the average yield per plant was determined to be 31.21 ± 11.42 g for one allele and 37.90 ± 15.38 g for the other allele of *OsCYP71P6-1*. This difference in yield was statistically significant, indicated by a *p* value of 0.03. 

The mean phenotypic difference between the two subpopulations identified in the STRUCTURE analysis exhibited significant distinctions for several traits including single-plant yield, number of spikelets, filled grains, and number of productive tillers. Notably, the single-plant yield trait displayed average values of 37.06 ± 15.33 g for subpopulation I and 31.42 ± 10.48 g for subpopulation II, with a significant *p* value of 0.03. Similarly, the mean number of spikelets was found to be 150.93 ± 28.71 and 185.32 ± 43.63 between the populations, respectively, with a *p* value of 0.02. In addition, the mean difference for the subpopulations along with admixtures also showed significant difference for single-plant yield, number of spikelets, and filled grains per panicle. Moreover, all the varieties were categorized into four distinct haplotypes based on the amplicon lengths of both primers. These groups were designated as follows: Group-1 (320 and 380 bp), Group-2 (350 and 380 bp), Group-3 (320 and 400 bp), and Group-4 (350 and 400 bp). Among these haplotypes, the single-plant yields observed were 29.81 ± 10.24 g, 37.71 ± 14.41 g, 36.64 ± 16.43 g, and 42.32 ± 11.73 g for the 1st, 2nd, 3rd, and 4th haplotypes, respectively, and the *p* value for the mean difference between the haplotypes was found to be highly significant (*p* = 0.005) (Figure 10, Table 5).

### 2.10. Association of OsCYP71P6 SNPs with the Spikelet Fertility Using Linear Regression Model

We studied the 572 SNPs including in/dels of the *OsCYP71P6* gene along with the promoter (Chr12:9577747.9584014) identified among the 1858 genotypes in 3K rice database for their association with spikelet fertility scores using linear regression models. The analysis showed eight SNPs (Table 6) had significant associations, with *p* values less than 0.05 for spikelet fertility (Chr12: 9581604; Chr12:9582455; Chr12:9582489; Chr12:9582557; Chr12:9582591; Chr12: 9582869; Chr12:9583776; Chr12:9583083; and Chr12:9582921).

Within the set of eight SNPs, a solitary SNP was detected in the coding region, leading to the Ser33Leu amino acid substitution. This change occurred in the span between the signal peptide and the p450 functional domain of the OsCYP71P6 protein. In contrast, all other SNPs were situated in the promoter region of *OsCYP71P6*. Furthermore, among these eight SNPs, six distinct haplotypes emerged. Notably, Hap2 exhibited a lower percentage (4.6%) of genotypes with a spikelet fertility score of 5, contrasting with the other haplotypes (11.9%). The specific sequence data for the *OsCYP71P6* haplotypes can be found in Appendix A. Moreover, Hap4 also displayed a reduced count of genotypes (0.8%), with a spikelet fertility score of 7, relative to the other haplotypes (1.7%). The variance analysis unveiled significant distinctions in the number of genotypes across different spikelet fertility scores among the haplotypes (*p* value: 0.00006) (refer to Appendix A).

## 3. Discussion

Serotonin is synthesized in plants from the amino acid tryptophan via decarboxylation to tryptamine, which is then hydroxylated to form serotonin via the shikimate pathway [27]. Tryptophan decarboxylase (TDC) and tryptamine-5-hydroxylase (T-5-H) catalyze these reactions [27]. *CYP71A1* is a major gene functioning in the serotonin pathway and is reported to be involved in biotic and abiotic stress tolerance in rice [27,45]. In this work, a total of 105 *OsCYP71* family genes were identified in the indica rice genome. These genes were further categorized into twelve distinct subfamilies. Notably, members sharing the same clade exhibited resemblances in terms of gene structures and conserved motifs. In addition, 105 *OsCYP71* genes were distributed on 11 chromosomes, and 36 pairs of *OsCYP71* involved in gene duplication events were found. The promoter region of *OsCYP71* contains a multitude of cis-elements associated with responses to light, hormones, and various stress conditions. Further, transcriptome profiling revealed that a majority of the genes within the OsCYP71 family were responsive to hormonal stimuli and exhibited induction across various tissues and developmental stages in rice. It was discovered that alleles of *OsCYP71P6* located in the promoter region play a role in regulating both spikelet fertility and yield-related traits in rice. Furthermore, a non-synonymous substitution adjacent to the signal peptide region of OsCYP71P6 and promoter in/dels were identified as having a notable association with spikelet fertility scores in rice. The implications and the significance of these findings are discussed in detail below. 

### 3.1. Identification and Evolutionary Analysis of the OsCYP71 Gene Family in Rice

The cytochrome P450 (CYP450) gene family plays a critical role in plant growth and developmental processes, nutrition, and detoxification of xenobiotics [1,2,3,5]. In this study, a total of 105 *OsCYP71* family genes were identified in the rice genome, compared with *Medicago truncatula* (59), *Arabidopsis thaliana* (54), *Vitis vinifera* (125), *Zea mays* (55), *Sorghum bicolor* (87), and *Glycine max* (53) [5,17,18,40,46,47]. This variation could potentially be attributed to gene duplication events. Phylogenetic relationships revealed that *OsCYP71* gene family members were divided into 12 subfamilies (Figure 1). Group VII encompassed a total of 74 members. Conversely, Groups I, II, III, IV, V, VI, VIII, IX, X, XI, and XII contained 0, 8, 5, 0, 13, 2, 3, 0, 0, 0, and 0 members, respectively (Appendix A). Based on the phylogenetic analysis, we assume that the *OsCYP71* family has a significant degree of evolutionary expansion in rice. Further, subcellular localization predicts that *OsCYP71* gene family members are mainly situated in the endomembrane system. The wide range of Mw and pI of *OsCYP71* may decide their functional diversity in different metabolic pathways (Table 1). 

Chromosome mapping revealed that the distribution of *OsCYP71* genes is uneven across a total of 11 chromosomes (Figure 2A). There are hot spots or gene clusters on Chr2, Chr3, Chr6, Chr8, Chr9, and Chr10 (Figure 2B). The *OsCYP71* gene was randomly and unevenly distributed on 12 chromosomes, and WGD or segmental duplication events were detected in each chromosome. Our results suggest that gene duplication affects *OsCYP71* chromosome location and gene family expansion depends on sequence duplication, either in WGD or segmental events. Apart from the segmental type, genome-wide duplication events also contribute to the alteration in the count of family members. In general, gene duplication plays an important role in the expansion and evolution of gene families, in which tandem repeats produce gene clusters or hotspots and fragment repeats produce homologous genes [48]. In this study, we identified a total of 36 instances of gene duplication within the *OsCYP71* gene family (Figure 3 and Appendix A). Further, different lengths of *OsCYP71* may play an important role in diversifying gene functions. The Ka/Ks score for these genes was observed to be less than one, suggesting a robust purifying selection with minor alterations following gene duplication. These findings provide evidence that the *OsCYP71* family genes have undergone a conserved evolutionary process. Furthermore, upon conducting an analysis of the exon-intron structure of *OsCYP71* genes, it became evident that the quantity of introns exhibited variation across distinct subfamilies (Figure 4). Across all CYP subfamily genes, the count of introns varies from zero to four. However, the majority of these genes typically possess between one and two introns. The maximum four introns were detected in *OsCYP71E2*, while *OsCYP71AD*, *OsCYP71C3*, *OsCYP71C4*, *OsCYP71AB7*, *OsCYP71W5*, *OsCYP71U3*, *OsCYP71T4*, *OsCYP71T7*, *OsCYP71T5*, *OsCYP71T8*, *OsCYP71T6*, and *OsCYP71T9* were intronless (Figure 4). Several studies have shown that a number of introns of CYP450 superfamily genes may have been lost during evolutionary processes in plants [13,49,50,51]. Additionally, a conserved motif analysis can offer deeper insights into the functional distinctions among family members. According to the findings from MEME, we identified the presence of motif 10 in OsCYP71 proteins. Motifs 1, 2, 3, 6, 7, and 10 were observed to exhibit strong conservation across the majority of OsCYP71 proteins (Figure 5A,B). The amino acid sequence alignment showed that OsCYP71 proteins consist of the conserved SRS and the heme-binding motifs (Appendix A). We also predicted the 3D structure of OsCYP71 proteins (Appendix A). Thus, based on these results, although certain motifs exhibit strong conservation within the OsCYP71 family, diverse subfamilies display unique motifs that potentially participate in specialized functions.

### 3.2. The OsCYP71 Gene Family Contains Multiple Cis-Regulatory Involved in Plant Developmental Processes

Cis-elements in the promoter region play a key role in the regulation of gene expression [44,52,53,54]. In this study, numerous cis-regulatory elements were identified within the promoter regions of *OsCYP71*. These encompassed elements responsive to phytohormones, light, cell cycle regulation, circadian rhythms, seed-specific regulation, as well as defense and stress responses (Figure 6A,B). In terms of hormonal response, ABRE, AuxRE, GARE, SARE, and MeJARE were the primary elements (Appendix A), and similar results have been observed in other species [12,13,50,51]. Many researchers demonstrated that gene duplication is a key factor in gene family expansion [1,55]. Gene duplication expanded the number of gene family members under evolutionary pressures; further, mutations in these genes may modulate the expression patterns of gene family members [56,57,58,59]. The CYP450 gene family holds a pivotal function in various aspects of plant growth and development, as well as in processes related to nutrition and the detoxification of xenobiotics in plants [1,3,4,5]. In addition, CYP450 has been implicated in diverse metabolic reactions and targets a wide range of biological molecules. These biosynthetic reactions lead to diverse plant hormones, fatty acid conjugates, lignin, secondary metabolites, and numerous defensive compounds [8,9]. It is intriguing to hypothesize that gene expression blueprints are evidence of their biological importance. The expression patterns of *OsCYP71* genes were thoroughly investigated across diverse tissues and developmental stages and under various hormone treatments. Our analysis revealed that the 105 *OsCYP71*-family genes studied exhibited differential expression across various tissues, developmental stages, and in response to hormone treatments (Figure 7A–C). For instance, *CYP71X4* and *CYP71X12* expression was elevated in ovary, embryo, and endosperm tissue, while the expression of *CYP71X13P* was induced in lemma, palea, and endosperm tissue. Further, the expression of *CYP71X7* was highly induced in vegetative, reproductive, ripening leaf blade, vegetative, reproductive leaf sheath, ovary, embryo and endosperm tissue, whereas the expression of *CYP71Z6*, *CYP71R2P*, *CYP71R1*, and *CYP71K12* was highly up-regulated in vegetative, reproductive, ripening leaf blade, and reproductive leaf sheath tissue. In addition, *CYP71Q1* expression was elevated in the shoot after 3 h, 6 h, and 12 h of the ABA and JA treatments, while the expression level of *CYP71Z4* was induced in the shoot after 3 h, 6 h, and 12 h of GA treatment. The transcript levels of *CYP71C16*, *CYP71Z6*, *CYP71C19P*, *CYP71V2*, *CYP71X10*, *CYP71W1*, and *CYP71T1* were induced in the shoot after 3 h, 6 h, and 12 h of JA treatment. Further, expression of *CYP71X10* increased in the shoot after 3 h, 6 h, and 12 h of BRS treatment, while *CYP71X11* was elevated in the shoot after auxin treatment for 1 h, 3 h, 6 h, and 12 h (Figure 7B). Following cytokinin treatment, the expression of *CYP71X12* increased in the root at the 1 h, 3 h, and 6 h time points. Conversely, under JA treatment, *CYP71X14*, *CYP71P1*, and *CYP71C16* were notably induced in the root after 30 min, 1 h, and 3 h. Furthermore, *CYP71Z4*, *CYP71W1*, *CYP71K5*, *CYP71Z2*, *CYP71C20*, and *CYP71Z3* were highly up-regulated in the root after JA treatment for 3 h and 6 h (Figure 7C). Similarly, the member of *BnCYP86* gene family exhibited expression in different tissues and responses to diverse environmental stresses, as did ABA in *Brassica Juncea* [13]. The levels of *SlCYP71AX* and *SlCYP77A20* gene expression were elevated in both the green and mature green stages of tomato fruit [50]. Collectively, these results have shown that *OsCYP71* gene family members may participate in different developmental processes and responses to phytohormones in rice. Thus, the outcome of this work may provide important insight into deciphering the biological functions of *OsCYP71* gene family members in future.

### 3.3. The Effect of Promoter In/Dels of OsCYP71P6 on Yield-Related Traits

The promoter in/del alleles of *OsCYP71P6* exhibited differences of 20.19%, 13.65%, 5.37%, 8.79%, and 36.86% in relation to single-plant yield, number of spikelets per panicle, panicle length, panicle weight, and unfilled grains per panicle, respectively, across various rice varieties. In support of these findings, the previous report by Lu et al. [28] showed the association of the loss-of-function allele of *OsCYP71P6* with the yield and number of tillers in rice. This suggests that the specific promoter allele of OsCYP71P6 could be correlated with the gene’s expression level, thereby playing a role in regulating yield-related traits in rice varieties developed for cultivation in India. Previously, it was discovered that promoter insertion/deletions (in/dels) within the malate transporter 9 (Al-MT9) gene in tomato were linked to variations in malate content [60]; the Arabidopsis Fumarase gene (FUM) promoter in/dels regulates carbon assimilation and nitrogen use [61]; the wheat promoter in/del in the elongation factor (TEF-7A) was found to be associated with the grain number per spike [62]. Promoter insertion/deletion (In/Dels) polymorphisms have been linked to heterotic gene expression in hybrid varieties [63]. The promoter in/dels in these examples resulted in altered gene expression and associated phenotypes. Thus, *OsCYP71P6* gene expression variation could be related to the identified promoter in/dels, which needs to be further confirmed. Furthermore, the *OsCYP71P6* gene participates in the serotonin biosynthesis pathway. Increasing the expression of tryptophan decarboxylase led to elevated serotonin levels. However, this led to a notable decrease in yield due to changes in rice panicle branching and the quantity of spikelets [64]. Additionally, overexpression of rice *OsSNAT1* (Serotonin N-Acetyl Transferase-1), which converts serotonin into melatonin, enhanced the number of panicles per plant and yield [65]. This supports the hypothesis that high levels of serotonin could have a negative effect on the yield. Therefore, it is feasible that elevated serotonin levels exert a negative regulatory effect on yield-related traits in rice varieties and targeting *OsCYP71P6* promoter insertions/deletions could be a potential strategy to finely adjust both gene expression and serotonin levels in rice, ultimately leading to increased yields. 

### 3.4. The Effect of 3′-UTR In/Dels of OsCYP71P6 on Single-Plant Yield

The 3′-untranslated region (3’-UTR) allele of *OsCYP71P6* also showed significant difference for the single-plant yield in rice varieties. The single-plant yield was 17.65% higher in the varieties having the insertion allele (350 bp) of the *OsCYP71P6-1* primer. Previously, it was reported in different genes and crops that sequence variation in the 3’-UTR region affects gene expression and was associated with major traits [66,67,68]. Therefore, variation in the 3′-UTR of *OsCYP71P6* might influence the expression or stability of the gene’s transcripts, subsequently impacting its related yield traits in rice.

### 3.5. A Non-Synonymous Substitution Near the Signal Peptide Region of the OsCYP71P6 Gene Regulates Spikelet Fertility

Haplotype 4 of *OsCYP71P6* exhibited a notably lower proportion of individuals (0.8%) with elevated spikelet fertility scores (SF7) compared to the other five haplotypes (1.61%). The SNPs leading to the amino acid substitution of Ser33Leu are situated close to the carboxy end of the trans-membrane (TM) signal peptide region within the OsCYP71P6 protein. This indicates that the substitution might be regulating the *OsCYP71P6* cellular localization. The eukaryotic CYPs are localized in the membrane, especially the endoplasmic reticulum, and are affected by the membrane properties [69]. Recently, it was found that mutations in the TM region affected the enzymatic function of CYPs [70]. Furthermore, a comparable region within closely related homologs of *OsCYP71P6* in rice also displayed sequence variations, including deletions or substitutions affecting serine amino acids (LOC_Os09g26940). This suggests that CYP71 homologs in rice exhibit substantial diversity in their signal peptide regions, and potentially in cellular localizations as well. Consequently, it becomes crucial to determine the functional importance of the amino acid substitution of Ser33Leu in relation to spikelet fertility. 

### 3.6. Gene Diversity of OsCYP71P6 in Rice Varieties

In this study, distance- and model-based analysis of in/dels in the *OsCYP71P6* gene grouped the rice varieties into two major groups. In similar findings, two sub-populations were also identified for the *Dense and Erect Panicle 1* (DEP1) gene in high-yielding japonica rice varieties [71]. Further, significant differences were observed in single-plant yield and the number of spikelets per panicle (17.6% and 14.1%, respectively) between the varieties categorized within the two distinct sub-populations. This underscores the correlation between genetic variations in the *OsCYP71P6* gene and yield-related traits in rice. Of particular significance is the identification of four in/del-based haplotypes within Indian rice varieties, which exhibit substantial differences in single-plant yield, emphasizing that the rice breeding initiative for varietal enhancement possesses ample diversity in the *OsCYP71P6* gene that can be strategically harnessed for genetic improvement. In support of this observation, two to three alleles were identified in the major yield-related genes (GS3, qSW5, GS5, Gn1a, and DEP1) in high-yielding varieties grown in Northern China [72]. However, the favorable allele proportion of the *OsCYP71P6* gene identified in this study was only ~3%. Therefore, targeted breeding for introgression of the favorable allele of *OsCYP71P6* could be attempted to increase the genetic gain for yield in rice. In this regard, the favorable haplotype (Hap4) of *OsCYP71P6* identified in four rice varieties (PR106, PR113, HPR2143, Himalaya-1) could be used as a donor for marker-assisted selection for yield improvement, whereas the most common haplotypes present in the rice varieties showed the least mean single-plant yields. Previously, utilizing sequence-based methodologies on major yield-related genes in rice varieties led to the discovery of novel alleles associated with traits related to yield [73]. In comparison, the gene-specific in/dels marker approach taken in this study could also be utilized for the marker-assisted breeding program in rice. Additionally, evaluation of the *OsCYP71P6* haplotypes in various rice varieties to determine their influence on traits related to both biotic and abiotic stress tolerance holds significant importance for the effectiveness of breeding efforts and genome editing programs [28]. Nowadays, the rice 3K panel, comprised of sequence information of ~3000 rice genotypes, is used for the identification of the favorable haplotypes of key genes in rice [74]. In this study, information from the 3k database was leveraged to create in/del primers. These primers were developed with the aim of comprehending the diversity of the *OsCYP71P6* gene among rice varieties that have been cultivated in India. A similar approach was used in our previous report to identify the UTR specific in/dels in the *TPP7* gene for tolerance to germination under submerged conditions [39]. Therefore, this strategy would be immensely valuable for uncovering gene diversity within key genes in the rice genome. Similarly, major gene-specific in/dels have been identified for the yield-related gene Gn1a in rice [68,75]. The superior haplotypes that have been identified could be incorporated into widely grown rice varieties using marker-assisted backcross breeding techniques.

## 4. Materials and Methods

### 4.1. Identification and Characterization of OsCYP71 Members in Rice Genome

A hidden Markov Model (HMM) profile of the OsCYP71 conserved domain (PF00067) was obtained from the Pfam database [76] by screening in the rice genome database. OsCYP71 family members were further identified through the NCBI-CDD [77] and SMART databases [78]. An isoelectric point calculator was utilized to analyze the theoretical isoelectric point (PI) and molecular weight (MW) of the OsCYP71 protein [79]. PSORT and BUSCA were used to predict the subcellular localization of OsCYP71 encoded proteins [80,81].

### 4.2. Phylogenetic Tree, Gene Structure, and Conserved Motif Analysis of the OsCYP71 Family in Rice

Arabidopsis, tomato, and OsCYP71 protein sequences were acquired from Ensembl Plants (https://plants.ensembl.org/index.html, (accessed on 20 June 2023)) and, subsequently, phylogenetic trees were constructed using MEGA 11 [81]. ClustalW was used for sequence alignment, the maximum likelihood method was used for phylogenetic tree construction, and the reliability of the tree obtained was determined using the bootstrap method with 1000 replicates. The exon-intron structure of OsCYP71 genes was visualized and mapped using the Gene Structure Display Server (GSDS) [82]. The conserved motifs of OsCYP71 protein sequences were plotted by the MEME webserver [83]. The Phyre2 web server was used to create the 3D structure of OsCYP71 [84].

### 4.3. Chromosome Localization, Gene Replication, Cis-Regulatory Elements, GO Enrichment, and Expression Analysis

The acquisition of chromosome localization information for *OsCYP71* genes was carried out through Ensembl Plants (http://plants.ensembl.org/biomart/martview, (accessed on 20 June 2023), aiming to facilitate their placement on the respective chromosomes. The *OsCYP71* gene family members were mapped using PhenoGram [85]. Gene replication and Ka/Ks values were analyzed using MCScan tools [86] and TBtools [87]. The 1500 bp sequence upstream of the OsCYP71 genes was used for promoter element analysis by the PlantCARE webserver [88]. GO enrichment of OsCYP71 proteins was performed using the singular enrichment analysis (SEA) function of a network-based agriGO program [89] Tissue-specific expression values for the 105 *OsCYP71* genes were extracted from the RiceXPro database (http://ricexpro.dna.affrc.go.jp, (accessed on 28 June 2023)) and, subsequently, heatmaps were generated using ClustVis [90].

### 4.4. Phenotyping for Different Traits

The study was carried out at ICAR National Rice Research Institute, Cuttack in January 2021. A total of 131 released rice varieties were considered for the phenotyping data analysis (Appendix A). Approximately a hundred seeds of the varieties were line sown in the nursery and transplanted in the puddled field conditions, with two lines per replication and each line consisting of fifteen seedlings. Additionally, two replications and alpha lattice designs [91] to reduce the variability in field experiments were maintained for the evaluation of the yield-related traits. The standard management practices of application of fertilizer and irrigation management were followed and at the physiological maturity stage, the number of productive tillers (nos.) was directly measured in three plants per line in the field. Further, at the crop’s maturity stage, various parameters including panicle length (in centimeters), panicle weight (in grams), spikelet count, both filled and unfilled grain quantities, seed weight (in grams), and single-plant yield (in grams) were assessed. This analysis involved the harvesting of three plants per line. The collected panicles were sun-dried for a duration of one week before being utilized for the assessment of panicle-related characteristics. The methodologies for measuring these traits were consistent with previously established protocols [92,93]. Briefly, panicle length was measured manually using a measuring scale from the top three panicles per plant. Further, the weight of these panicles was measured using a weighing machine. Additionally, the top three panicles per plant were also used for manually counting the number of spikelets per panicle along with filled and unfilled grains. Then, a hundred grains per line were manually counted and weighed for 100 g seed weight. Furthermore, the panicles of individual plants were subjected to threshing, and the filled spikelets were weighed to determine the yield per single plant. In parallel, young leaves were gathered for DNA extraction, rapidly frozen using liquid nitrogen, and subsequently preserved at a temperature of −80 °C for future investigations. 

### 4.5. Development of Gene Specific Polymorphic Variants-Insertion/Deletions (GPV-In/Dels)

Insertion/deletions present in the *OsCYP71P6* (CYP71A1, LOC_Os12g16720) gene, including the promoter, were identified from the SNP Seek database of rice (https://snp-seek.irri.org/, (accessed on 29 June 2023)). The in/del variants, present in approximately 3000 genotypes within the SNP Seek database, were detected by aligning against the Nipponbare reference genome. These variants were specific to certain genes. As a result of this development, the marker was designated as “Gene-Specific Polymorphic Variants-In/Dels” (GPV-In/Dels). Further, the CYP71A1 gene sequence of Nipponbare was retrieved from the RGAP database (http://rice.plantbiology.msu.edu/, (accessed 29 June 2023)) and utilized for designing the primers flanking the selected in/dels of size > 10 bp. The details of the primer sequence are given in Appendix A. In addition, the SNPs and in/dels, totaling around 572, that were discovered within the *OsCYP71P6* gene were subjected to analysis along with the spikelet fertility quality scores (ranging from 1 to 7) available in the 3 K rice database. This analysis involved investigating the associations between variants and traits through a linear regression model. The ‘lm’ function within the R software, specifically R Version 3.6.0, was employed for this purpose [94].

### 4.6. Experimental Validation of OsCYP71P6 In/Dels

Samples of leaves from various rice varieties were gathered to validate the newly created in/del markers. The CTAB (cetyl trimethyl ammonium bromide) technique was employed to extract genomic DNA from these leaf samples [92]. Using agarose gel electrophoresis, the purity of the DNA was determined (0.8 percent). For PCR amplification, genomic DNA was diluted with nuclease-free water to a working concentration (50 ng/µL). Initially, all four gene-specific in/del markers were employed to genotype rice varieties, facilitating an analysis of genetic diversity. The PCR was carried out in a 10 μL reaction volume containing 1 μL of template DNA, 1 μL of 10× buffer (1.5 mM Mg in 1×), 1.0 μL of dNTP (2.5 mM), 1 μL of each in/del primer (0.2 μM), and 0.2 U of *Taq* polymerase (Kapa Biosystem, Cape Town, South Africa), while the volume was made using double-distilled H_2_O. The PCR program includes an initial denaturation for 3 min at 95 °C, which was followed by 35 cycles of denaturation for 30 s at 95 °C, annealing for 45 s at 55 °C, and an extension for 1 min at 72 °C, with a final extension for 10 min at 72 °C using a thermal master cycler (Eppendorf, Hamburg, Germany). The PCR products were resolved using a 4.0 percent agarose gel and recorded using a UV gel documentation system (Bio-Rad, Hercules, CA, USA) (Appendix A). The alleles of the in/del marker were assessed manually, taking into account their amplicon length across various rice cultivars.

### 4.7. Diversity Analysis

The alleles identified in 131 rice varieties for *OsCYP71P6* gene in/del markers were used for gene diversity, cluster analysis, sub-population structures, and analysis of molecular variance (AMOVA). Gene diversity and cluster analysis were performed using power marker software [95]. Further, AMOVA was performed using the GenALEx V 6.5 tool using amplicon length allelic data as variables [96]. The STRUCTURE software was utilized to analyze population sub-structure, while Structure Harvester was employed to ascertain the optimal number for interpreting the results [97].

## 5. Conclusions

In this study, we identified 105 *OsCYP71* genes within the indica rice genome. These genes were subsequently classified into 12 distinct subfamilies based on shared characteristics. Notably, genes within the same subfamily exhibited similar gene structures and conserved motifs. Additionally, the distribution of these 105 *OsCYP71* genes spanned across 11 chromosomes, with 36 instances of *OsCYP71* gene duplication events. The promoter regions of *OsCYP71* genes were found to harbor a substantial number of cis-elements associated with light responsiveness, hormone signaling (including Auxin, cytokinin, GA, ABA, MeJA, JA, BRS, and SA), and various stresses such as drought and low temperature. Transcriptome profiling further revealed that a majority of genes within this family demonstrated responsiveness to hormones and were induced across diverse tissues and developmental stages in rice. Employing linear regression models, we identified eight promoters along with a gene variant (Ser33Leu) within *OsCYP71P6* that exhibited a significant association with spikelet fertility. Furthermore, the allelic effects of different *OsCYP71P6* alleles, identified through in/dels polymorphism in 131 rice varieties, were validated for their impact on yield-related traits. Our investigations also revealed that the *OsCYP71P6* gene plays a role in the regulation of spikelet count, filled grains, single-plant yield, panicle length, and panicle weight in various rice varieties. These findings serve as a robust foundation for deeper exploration of the functions of *OsCYP71*-family genes in a wide array of biological processes. Additionally, the outcomes of our study underscore the potential influence of promoter allelic variation and the Ser33Leu amino acid substitution within the *OsCYP71P6* gene on yield-related traits in rice. As a result, the promoter variants of *OsCYP71P6* that we have identified hold promise for utilization in efforts aimed at enhancing rice yield.

## Figures and Tables

**Figure 1 plants-12-03035-f001:**
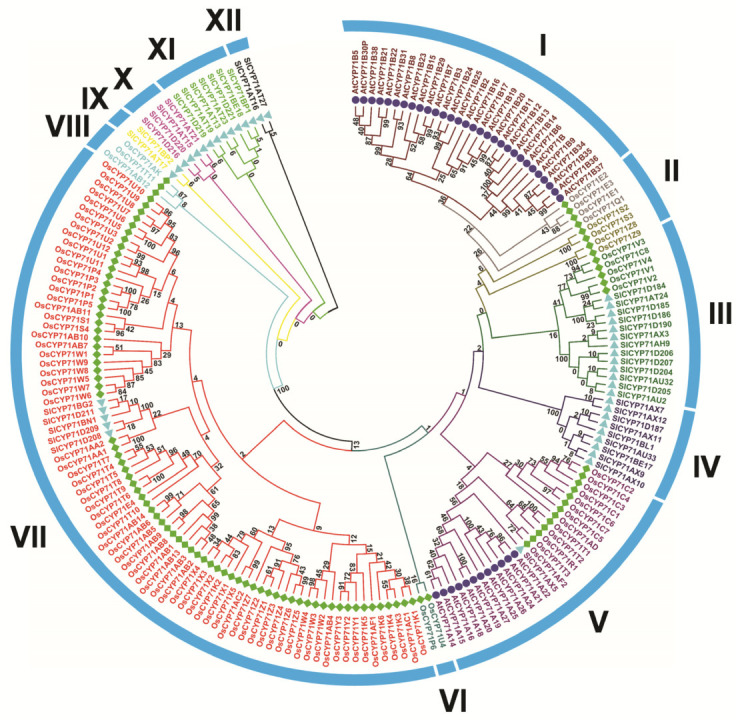
The phylogenetic tree of CYP proteins was established for *A. thaliana*, *O. sativa*, and *S. lycopersicum* using the neighbor-joining method in MEGA 11. To gauge the tree’s reliability, 1000 bootstrap replicates were employed. *OsCYP71* genes could be divided into 12 subfamilies (I–XII).

**Figure 2 plants-12-03035-f002:**
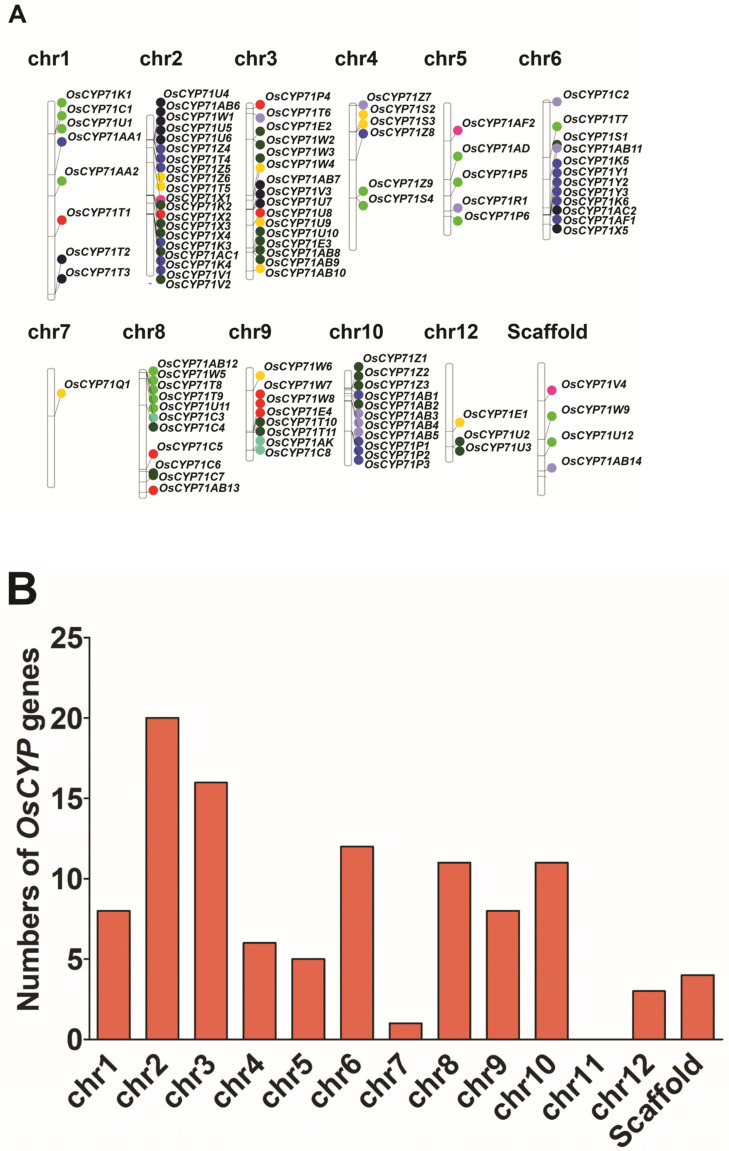
Chromosome distribution of identified *OsCYP71* genes. (**A**) Schematic illustrations of the chromosomal distribution of CYP genes on the twelve chromosomes of rice with the gene positions on each chromosome represented by a line on the right side. (**B**) Dispersal of CYP71 genes across eleven chromosomes of rice.

**Figure 3 plants-12-03035-f003:**
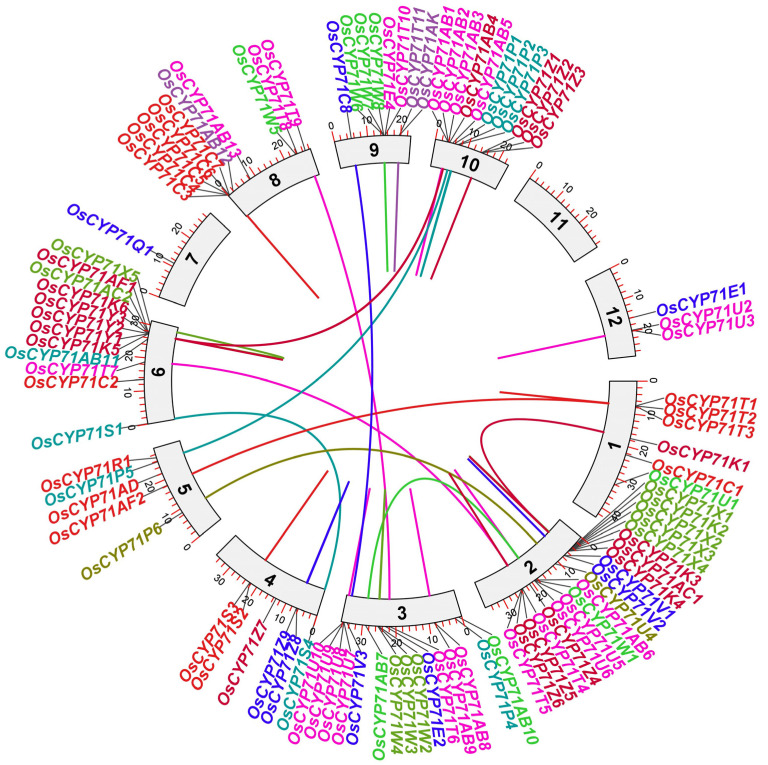
Chromosomal distribution and duplicated CYP gene pairs in rice. Duplicated CYP gene pairs are connected with distinct colors of lines. The figure was constructed using TB tools with numbers 1–12 denoting the rice chromosomes.

**Figure 4 plants-12-03035-f004:**
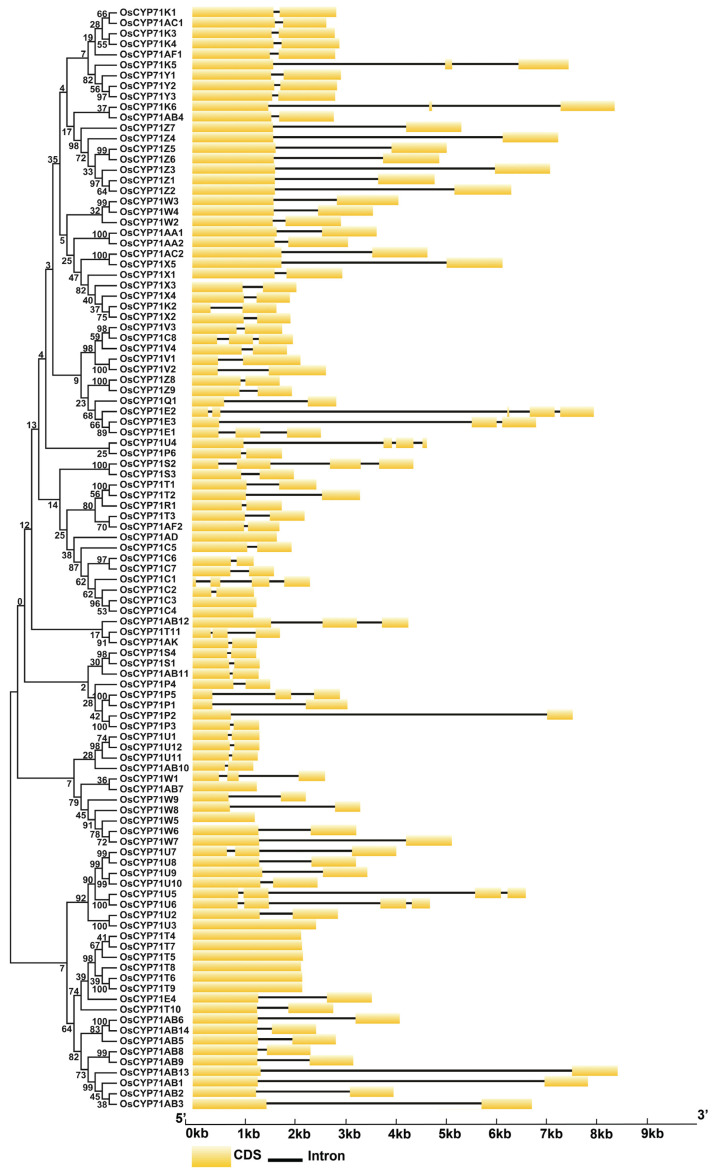
The intron-exon structure of the *OsCYP71* genes. The yellow boxes signify exons while black lines denote introns. The lengths of the boxes and lines are scaled based on gene length.

**Figure 5 plants-12-03035-f005:**
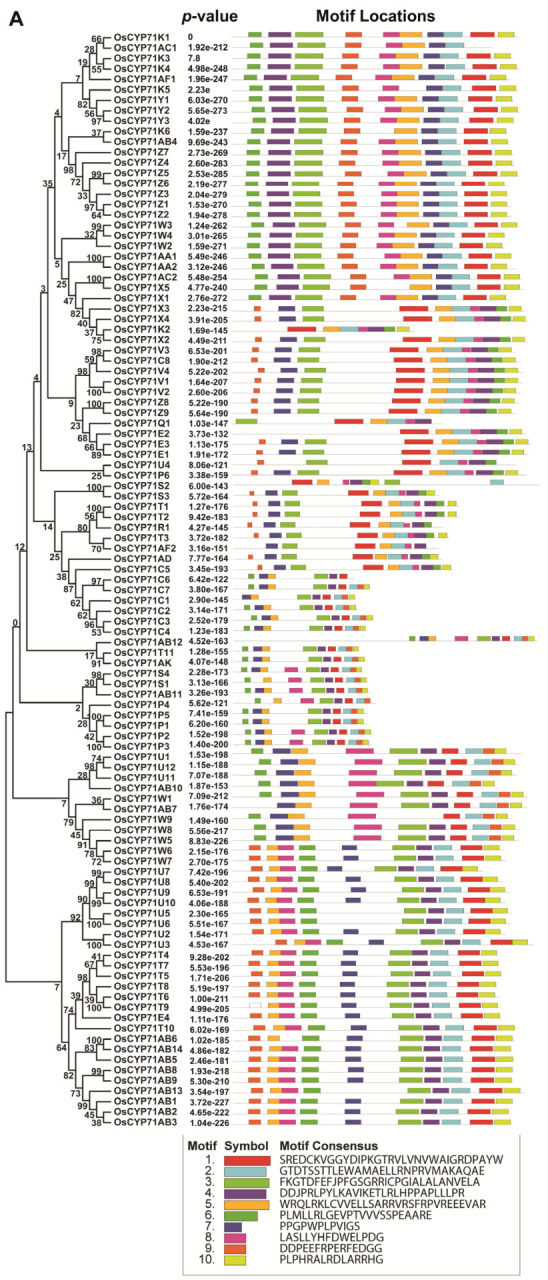
The conserved motifs in OsCYP71 genes. The conserved motifs were elucidated by the MEME database. (**A**) The distinct colored boxes represent different conserved motifs with variable sizes and sequences. (**B**) Sequence logo of the conserved motif of OsCYP71.

**Figure 6 plants-12-03035-f006:**
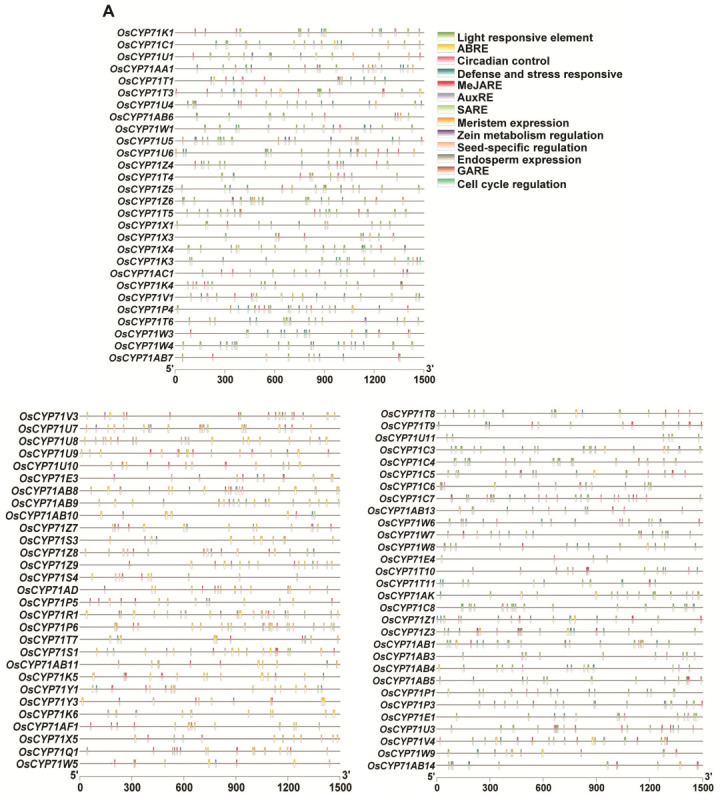
Identification of cis-regulatory elements in the 1500 bp promoter region of the *OsCYP71* gene family. (**A**) Phytohormone-responsive elements, light-responsive elements, growth- and development-related elements, stress-responsive elements and other elements with unknown functions are shown by distinct colors. (**B**) The numbers of different CAREs found in *OsCYP71* gene family members.

**Figure 7 plants-12-03035-f007:**
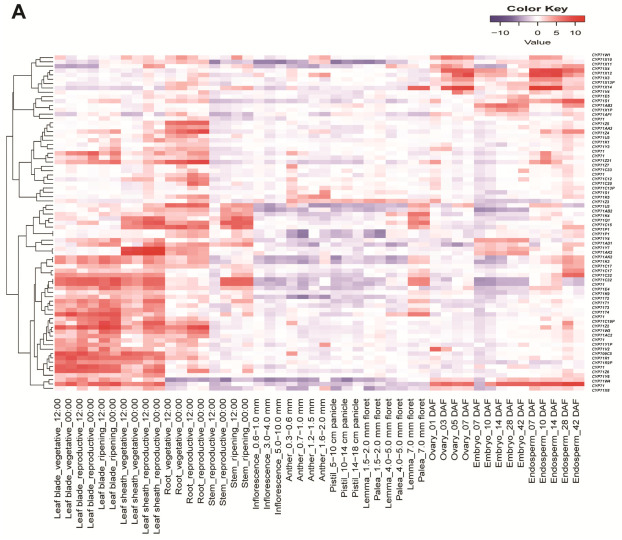
Expression profiles of *OsCYP71* genes in different tissues and hormone treatments. (**A**) The expression profile in various tissues and developmental stages. (**B**) The expression profile in shoots under different hormone treatments. (**C**) The expression profile in roots under different hormone treatments.

**Figure 8 plants-12-03035-f008:**
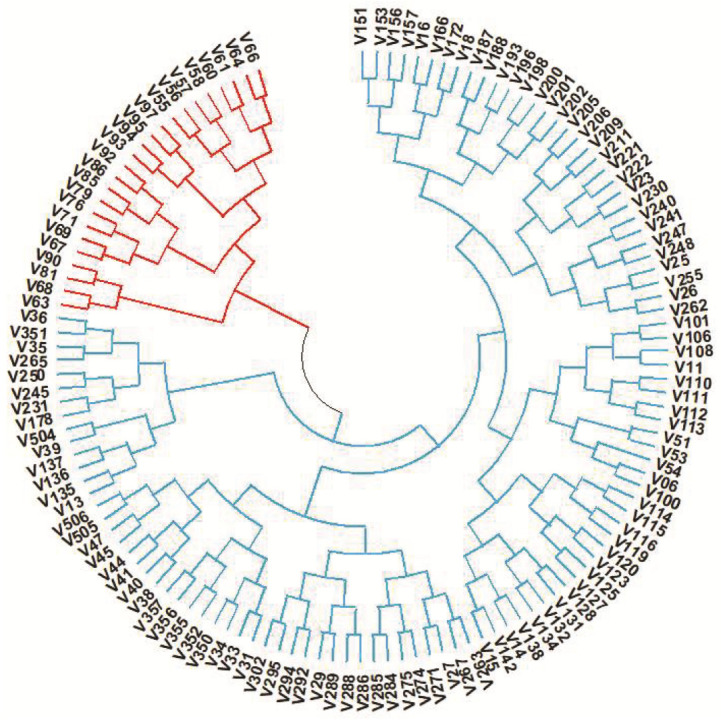
Phylogenetic analysis of rice varieties through the unweighted pair group method with arithmetic averaging (UPGMA) model using in/dels allelic variants of *OsCYP71P6*. Cluster A and Cluster B indicate major clusters formed using distance-based diversity analysis. V—Varieties. The names of the varieties are given in Appendix A.

**Figure 9 plants-12-03035-f009:**
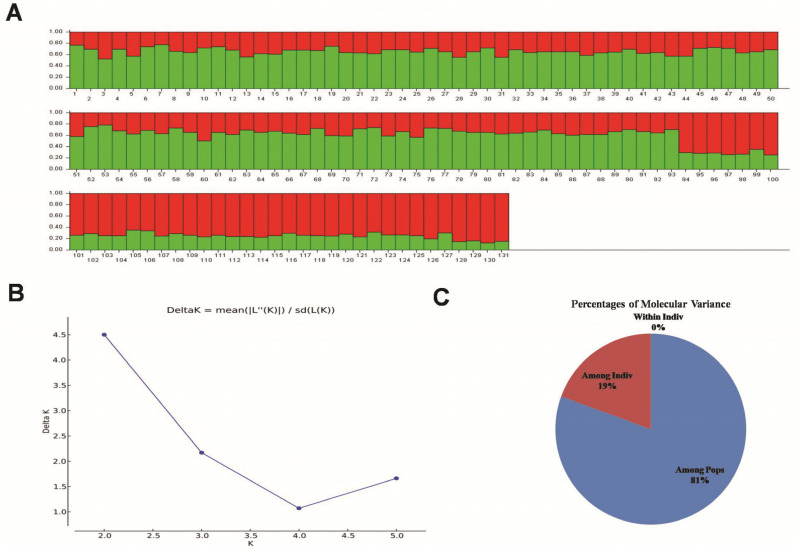
Model-based diversity analysis and analysis of molecular variance (AMOVA) for sub-populations. (**A**) Sub-structure analysis of rice varieties using the allelic variants of *OsCYP71P6*. (**B**) Delta K population structure plot for the rice varieties produced through the STRUCTURE harvester program. (**C**) Pie chart of the percent distribution of the analysis of molecular variance (AMOVA) for the *OsCYP71P6* alleles in rice varieties.

**Figure 10 plants-12-03035-f010:**
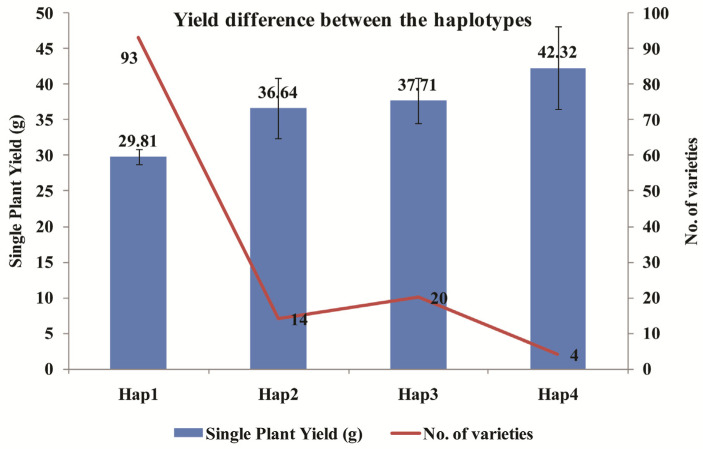
Yield differences between the four haplotypes of *OsCYP71P6* allelic variants. Hap 1-4 indicates the four different haplotypes of *OsCYP71P6-1* and *OsCYP71P6-4* alleles. No. of varieties present in each haplotype is represented by a line graph. Standard error was used to plot the error bar in the bar chart. The number indicates the mean yield and no. of varieties for the haplotypes. Statistical difference in the mean of the single-plant yield between the haplotypes was analyzed using the Z test for mean difference. *p* = 0.005 represents the statistically significant mean difference value calculated using the Z Test at a 1% level of significance. Hap1: *OsCYP71P6-1*_320bp_, *OsCYP71P6-4*_380bp_, Hap2: *OsCYP71P6-1*_350bp_, *OsCYP71P6-4*_380bp_, Hap3: *OsCYP71P6-1*_320bp_, *OsCYP71P6-400*_bp_, Hap4: *OsCYP71P6-1*_350bp_, *OsCYP71P6-400*_bp_, bp-base pair (amplicon size).

**Table 1 plants-12-03035-t001:** The identified putative cytochrome P450 (*OsCYP71*) genes in rice and their biophysical characteristics.

Proposed Gene Name	Gene ID	Chromosome	Genomic Location	Orientation	CDS Length (bp)	Protein Length (aa)	Molecular Weight (KDa)	Isoelectric Point (pI)	GRAVY	Predicted Subcellular Localization
*OsCYP71K1*	BGIOSGA001610	1	1:17410528-17412154	Reverse	1557	518	57.41	7.73	0.0144	endomembrane system
*OsCYP71C1*	BGIOSGA003715	1	1:22500639-22503588	Forward	1413	470	53	5.85	−0.172	plasma membrane
*OsCYP71U1*	BGIOSGA004274	1	1:32296506-32298186	Forward	1584	527	58.27	8.09	−0.092	endomembrane system
*OsCYP71AA1*	BGIOSGA005209	1	1:46261115-46263199	Forward	1572	523	58.12	8.31	0.007	endomembrane system
*OsCYP71AA2*	BGIOSGA005210	1	1:46266756-46268515	Forward	1605	534	59.45	8.35	−0.029	endomembrane system
*OsCYP71T1*	BGIOSGA003049	1	1:7516842-7519131	Forward	1689	562	60.51	5.64	0.051	endomembrane system
*OsCYP71T2*	BGIOSGA003050	1	1:7524389-7527482	Forward	1692	563	60.53	6.62	0.019	endomembrane system
*OsCYP71T3*	BGIOSGA003054	1	1:7558245-7560317	Forward	1614	537	59.18	7.42	−0.018	endomembrane system
*OsCYP71U4*	BGIOSGA007962	2	2:11083073-11087397	Forward	1503	500	54.15	6.47	−0.027	endomembrane system
*OsCYP71AB6*	BGIOSGA008255	2	2:18910371-18913246	Forward	1518	505	56.15	8.88	−0.098	endomembrane system
*OsCYP71W1*	BGIOSGA008261	2	2:19021940-19025270	Forward	1614	537	60.1	8.70	−0.063	endomembrane system
*OsCYP71U5*	BGIOSGA008265	2	2:19083433-19088060	Forward	1590	529	59.11	8.17	−0.146	endomembrane system
*OsCYP71U6*	BGIOSGA008266	2	2:19089067-19092362	Forward	1578	525	58.86	8.43	−0.168	endomembrane system
*OsCYP71Z4*	BGIOSGA006339	2	2:20885281-20889418	Reverse	1542	513	56.9	7.98	−0.02	plasma membrane
*OsCYP71T4*	BGIOSGA008463	2	2:23403404-23404909	Forward	1506	501	55.31	8.00	0.032	endomembrane system
*OsCYP71Z5*	BGIOSGA008468	2	2:23502957-23505833	Forward	1566	521	57.34	8.22	−0.004	endomembrane system
*OsCYP71Z6*	BGIOSGA006215	2	2:23525585-23528377	Reverse	1557	518	57.46	7.51	−0.028	endomembrane system
*OsCYP71T5*	BGIOSGA006210	2	2:23642185-23643714	Reverse	1530	509	56.07	7.03	−0.009	endomembrane system
*OsCYP71X1*	BGIOSGA007002	2	2:5495789-5497484	Reverse	1560	519	57.57	7.63	−0.07	endomembrane system
*OsCYP71K2*	BGIOSGA007683	2	2:5517065-5518619	Forward	966	321	36.35	5.45	−0.089	endomembrane system
*OsCYP71X2*	BGIOSGA007686	2	2:5560619-5562432	Forward	1566	521	57.65	6.82	−0.031	endomembrane system
*OsCYP71X3*	BGIOSGA007688	2	2:5569750-5571671	Forward	1548	515	58.05	7.74	−0.131	endomembrane system
*OsCYP71X4*	BGIOSGA007691	2	2:5604868-5606669	Forward	1566	521	57.49	6.78	−0.042	endomembrane system
*OsCYP71K3*	BGIOSGA007694	2	2:5642839-5644448	Forward	1530	509	56.32	8.60	0.057	endomembrane system
*OsCYP71AC1*	BGIOSGA007695	2	2:5644996-5646510	Forward	1422	473	52.17	8.18	−0.055	endomembrane system
*OsCYP71K4*	BGIOSGA007696	2	2:5647245-5648907	Forward	1572	523	57.78	7.54	−0.052	endomembrane system
*OsCYP71V1*	BGIOSGA007791	2	2:7497274-7499270	Forward	1536	511	56.6	8.06	0.006	plasma membrane
*OsCYP71V2*	BGIOSGA007792	2	2:7500786-7503251	Forward	1527	508	56.49	7.97	0.057	endomembrane system
*OsCYP71P4*	BGIOSGA011508	3	3:2163933-2165882	Reverse	1644	547	59.2	6.95	0.047	endomembrane system
*OsCYP71T6*	BGIOSGA010447	3	3:21911994-21913514	Reverse	1521	506	55.75	6.45	0.009	endomembrane system
*OsCYP71E2*	BGIOSGA012992	3	3:23090126-23097523	Forward	1554	517	57.42	7.58	−0.409	plasma membrane
*OsCYP71W2*	BGIOSGA013057	3	3:25210900-25212580	Forward	1536	511	57.5	7.69	−0.096	endomembrane system
*OsCYP71W3*	BGIOSGA013059	3	3:25254727-25257057	Forward	1614	537	60.76	7.42	−0.094	endomembrane system
*OsCYP71W4*	BGIOSGA013063	3	3:25338238-25340280	Forward	1542	513	57.48	7.67	−0.063	endomembrane system
*OsCYP71AB7*	BGIOSGA010140	3	3:29073720-29075333	Reverse	1614	537	57.39	7.98	0.06	endomembrane system
*OsCYP71V3*	BGIOSGA013536	3	3:34645395-34647054	Forward	1506	501	55.22	9.30	0.034	endomembrane system
*OsCYP71U7*	BGIOSGA013602	3	3:35634216-35637043	Forward	1425	474	52.31	7.62	−0.047	endomembrane system
*OsCYP71U8*	BGIOSGA013604	3	3:35641266-35643533	Forward	1542	513	56.61	7.74	−0.012	endomembrane system
*OsCYP71U9*	BGIOSGA013605	3	3:35649465-35651890	Forward	1584	527	58.39	7.46	−0.07	endomembrane system
*OsCYP71U10*	BGIOSGA013606	3	3:35654380-35656113	Forward	1557	518	57.89	8.64	−0.033	endomembrane system
*OsCYP71E3*	BGIOSGA013948	3	3:40194199-40200533	Forward	1584	527	58.42	9.44	−0.028	endomembrane system
*OsCYP71AB8*	BGIOSGA011112	3	3:8385202-8386836	Reverse	1506	501	55.39	8.28	−0.033	endomembrane system
*OsCYP71AB9*	BGIOSGA011111	3	3:8394147-8396376	Reverse	1503	500	56.22	8.12	−0.15	endomembrane system
*OsCYP71AB10*	BGIOSGA011586	3	3:954097-955621	Reverse	1446	481	52.38	7.52	0.105	endomembrane system
*OsCYP71Z7*	BGIOSGA016146	4	4:13300800-13303842	Forward	1536	511	56.74	8.24	0.0001	endomembrane system
*OsCYP71S2*	BGIOSGA014867	4	4:22441383-22445458	Reverse	2301	766	84.89	8.42	−0.324	endomembrane system
*OsCYP71S3*	BGIOSGA014866	4	4:22450395-22452270	Reverse	1536	511	55.88	8.52	−0.074	endomembrane system
*OsCYP71Z8*	BGIOSGA015504	4	4:6445565-6447178	Reverse	1524	507	56.65	6.89	−0.014	endomembrane system
*OsCYP71Z9*	BGIOSGA015981	4	4:6553171-6555010	Forward	1506	501	55.76	6.04	0.011	endomembrane system
*OsCYP71S4*	BGIOSGA015743	4	4:89164-90761	Reverse	1500	499	55.32	7.46	−0.05	endomembrane system
*OsCYP71AF2*	BGIOSGA019698	5	5:18144104-18145713	Forward	1536	511	55.4	9.54	−0.096	endomembrane system
*OsCYP71AD*	BGIOSGA018026	5	5:22038254-22039816	Reverse	1563	520	57.1	6.35	−0.087	endomembrane system
*OsCYP71P5*	BGIOSGA020098	5	5:25741450-25745152	Forward	1539	512	58.36	8.73	−0.237	endomembrane system
*OsCYP71R1*	BGIOSGA020185	5	5:27004831-27006483	Forward	1569	522	56.66	6.64	−0.083	endomembrane system
*OsCYP71P6*	BGIOSGA018523	5	5:8791390-8793045	Reverse	1560	519	57.32	7.62	−0.067	mitochondrial membrane
*OsCYP71C2*	BGIOSGA022809	6	6:13456419-13457960	Forward	1416	471	53.44	8.78	0.002	endomembrane system
*OsCYP71T7*	BGIOSGA021189	6	6:18739993-18741510	Reverse	1518	505	55.67	7.52	0.049	endomembrane system
*OsCYP71S1*	BGIOSGA022122	6	6:190381-192067	Forward	1557	518	56.41	8.18	0.007	endomembrane system
*OsCYP71AB11*	BGIOSGA020890	6	6:25633293-25634951	Reverse	1584	527	57.95	8.12	−0.085	endomembrane system
*OsCYP71K5*	BGIOSGA023340	6	6:27270045-27274299	Forward	1560	519	57.07	8.65	−0.042	endomembrane system
*OsCYP71Y1*	BGIOSGA023341	6	6:27281079-27282759	Forward	1539	512	55.46	7.93	0.089	endomembrane system
*OsCYP71Y2*	BGIOSGA023345	6	6:27304346-27305982	Forward	1569	522	57.41	7.70	−0.039	endomembrane system
*OsCYP71Y3*	BGIOSGA023346	6	6:27308296-27309910	Forward	1545	514	56.39	7.86	−0.008	endomembrane system
*OsCYP71K6*	BGIOSGA023350	6	6:27362563-27367339	Forward	1503	500	56.9	9.67	−0.053	endomembrane system
*OsCYP71AC2*	BGIOSGA023351	6	6:27374980-27377638	Forward	1632	543	60.63	7.55	−0.133	endomembrane system
*OsCYP71AF1*	BGIOSGA020805	6	6:27378645-27380259	Reverse	1515	504	55.24	6.38	−0.001	endomembrane system
*OsCYP71X5*	BGIOSGA020704	6	6:29562437-29565945	Reverse	1641	546	60.56	8.19	−0.125	endomembrane system
*OsCYP71Q1*	BGIOSGA024456	7	7:11074316-11076970	Reverse	1113	370	42.15	5.61	−0.053	endomembrane system
*OsCYP71AB12*	BGIOSGA027933	8	8:1799581-1805006	Forward	3498	1165	130.4	7.42	−0.086	mitochondrial membrane
*OsCYP71W5*	BGIOSGA026880	8	8:23757559-23759121	Reverse	1563	520	58.35	7.73	−0.038	endomembrane system
*OsCYP71T8*	BGIOSGA028809	8	8:24254355-24255857	Forward	1503	500	55.99	8.22	0.037	endomembrane system
*OsCYP71T9*	BGIOSGA026711	8	8:26774318-26775838	Reverse	1521	506	56.35	7.19	−0.012	endomembrane system
*OsCYP71U11*	BGIOSGA026530	8	8:29302265-29303901	Reverse	1554	517	56.33	6.26	−0.008	endomembrane system
*OsCYP71C3*	BGIOSGA027793	8	8:327857-329461	Reverse	1605	534	59.94	6.63	−0.003	plasma membrane
*OsCYP71C4*	BGIOSGA027791	8	8:335792-337318	Reverse	1527	508	57.55	6.70	−0.012	endomembrane system
*OsCYP71C5*	BGIOSGA027790	8	8:341090-342922	Reverse	1650	549	62.01	6.04	−0.192	endomembrane system
*OsCYP71C6*	BGIOSGA027840	8	8:347325-348858	Forward	1389	462	52.15	7.86	−0.084	endomembrane system
*OsCYP71C7*	BGIOSGA027841	8	8:358473-360517	Forward	1575	524	59.22	8.46	−0.0001	endomembrane system
*OsCYP71AB13*	BGIOSGA027437	8	8:8239548-8245451	Reverse	1575	524	58.18	9.90	−0.082	endomembrane system
*OsCYP71W6*	BGIOSGA030841	9	9:14907658-14909929	Forward	1545	514	58.11	8.71	−0.125	endomembrane system
*OsCYP71W7*	BGIOSGA030842	9	9:14912617-14916216	Forward	1557	518	58.96	7.81	−0.107	endomembrane system
*OsCYP71W8*	BGIOSGA029682	9	9:14918713-14922927	Reverse	1569	522	59.65	8.50	−0.208	endomembrane system
*OsCYP71E4*	BGIOSGA029664	9	9:15258631-15261118	Reverse	1533	510	55.49	7.01	0.003	endomembrane system
*OsCYP71T10*	BGIOSGA029663	9	9:15266334-15268285	Reverse	1518	505	54.62	7.39	0.031	endomembrane system
*OsCYP71T11*	BGIOSGA031134	9	9:19550851-19553047	Forward	1446	481	52.45	6.58	−0.002	endomembrane system
*OsCYP71AK*	BGIOSGA031135	9	9:19556511-19558129	Forward	1521	506	54.63	8.97	−0.012	endomembrane system
*OsCYP71C8*	BGIOSGA030097	9	9:5147576-5149433	Reverse	1530	509	56.53	8.78	−0.032	endomembrane system
*OsCYP71Z1*	BGIOSGA031844	10	10:14240993-14243733	Reverse	1572	523	58.4	7.21	−0.019	endomembrane system
*OsCYP71Z2*	BGIOSGA031843	10	10:14247493-14251099	Reverse	1575	524	58.4	7.80	−0.098	endomembrane system
*OsCYP71Z3*	BGIOSGA031842	10	10:14257972-14262017	Reverse	1560	519	57.9	8.56	−0.119	endomembrane system
*OsCYP71AB1*	BGIOSGA032580	10	10:3806982-3812472	Forward	1506	501	56.15	8.03	−0.045	endomembrane system
*OsCYP71AB2*	BGIOSGA032583	10	10:3912905-3915694	Forward	1485	494	55.26	8.18	−0.047	endomembrane system
*OsCYP71AB3*	BGIOSGA032584	10	10:3947165-3951250	Forward	1497	498	55.47	9.16	−0.038	endomembrane system
*OsCYP71AB4*	BGIOSGA032599	10	10:4268768-4270367	Forward	1512	503	57.9	6.60	−0.168	endomembrane system
*OsCYP71AB5*	BGIOSGA032634	10	10:5163945-5165932	Forward	1509	502	55.78	9.24	0.024	endomembrane system
*OsCYP71P1*	BGIOSGA032653	10	10:5794451-5798343	Forward	1542	513	57.9	6.92	−0.155	endomembrane system
*OsCYP71P2*	BGIOSGA032680	10	10:6875415-6884974	Forward	1608	535	58.49	6.56	0.02	endomembrane system
*OsCYP71P3*	BGIOSGA032683	10	10:7021038-7022711	Forward	1581	526	57.52	7.67	−0.072	endomembrane system
*OsCYP71E1*	BGIOSGA036106	12	12:16043912-16046285	Reverse	1569	522	58.42	8.83	−0.166	endomembrane system
*OsCYP71U2*	BGIOSGA035935	12	12:19695567-19697582	Reverse	1557	518	55.44	6.75	0.149	endomembrane system
*OsCYP71U3*	BGIOSGA035934	12	12:19701585-19703297	Reverse	1713	570	60.72	7.15	0.197	endomembrane system
*OsCYP71V4*	BGIOSGA037944	Scaffold	AAAA02035682.1:5479-7227	Forward	1542	513	57.3	7.36	−0.012	endomembrane system
*OsCYP71W9*	BGIOSGA038041	Scaffold	AAAA02036020.1:14706-17551	Forward	1530	509	57.07	8.21	−0.305	endomembrane system
*OsCYP71U12*	BGIOSGA038175	Scaffold	AAAA02036741.1:10396-12075	Reverse	1566	521	57.9	8.24	−0.071	endomembrane system
*OsCYP71AB14*	BGIOSGA038318	Scaffold	AAAA02037602.1:3877-5590	Forward	1509	502	56.24	9.26	−0.25	endomembrane system

ID: identity; bp: base pair; aa: amino acids; pI: isoelectric point; MW: molecular weight; GRAVY: grand average of hydropathy; KDa: Kilo dalton.

**Table 2 plants-12-03035-t002:** Analysis of gene diversity for the two *OsCYP71P6* in/del markers across various rice varieties.

Marker	Major Allele Frequency	No. of Varieties	Allele No	Gene Diversity	Heterozygosity	PIC
CYP71P6-1	0.8626	131	2.0000	0.2370	0	0.2090
CYP71P6-4	0.8168	131	2.0000	0.2993	0	0.2545
Mean	0.8397	131	2.0000	0.2682	0	0.2317

**Table 3 plants-12-03035-t003:** Analysis of molecular variance (AMOVA) for two-subpopulations and admixtures of the *OsCYP71P6* in/dels variants in rice varieties.

Source of Variation	df	SS	MS	Est. Var.	Percent Variation (%)
Among Populations	1	39.256	39.256	0.498	81%
Among Individual varieties	129	31.003	0.240	0.120	19%
Within Individual varieties	131	0.000	0.000	0.000	0%
Total	261	70.260		0.618	100%

**Table 4 plants-12-03035-t004:** Descriptive statistics analysis of in/dels variants of the *OsCYP71P6* gene for yield-related traits in rice varieties.

Trait	Mean				Median				Mode				Kurtosis				Skewness			
Primer	*OsCYP71P6-1*		*OsCYP71 P6-4*		*OsCYP71 P6-1*		*OsCYP71 P6-4*		*OsCYP71 P6-1*		*OsCYP71 P6-4*		*OsCYP71 P6-1*		*OsCYP71 P6-4*		*OsCYP71 P6-1*		*OsCYP71 P6-4*	
OsCYP71P6																				
Amplicon length (bp)	320	350	380	400	320	350	380	400	320	350	380	400	320	350	380	400	320	350	380	400
No. of tillers (Nos.)	10.29	11.98 ^a^	10.32	11.41	10	10.33	10	10.66	12	10.33	12	16.66	−0.08	2.19	1.62	−0.68	0.43	1.2	0.82	0.25
Panicle length (cm)	26.15	26.24	25.89	27.36 ^b^	26.2	25.96	25.9	27.21	26.46	_	26.46	_	0.78	−0.91	0.65	−0.08	−0.32	0.33	−0.27	0.41
Single-plant yield (g)	31.21	37.9 ^a^	30.71	38.48 ^b^	30.06	34.7	29.8	36.26	33.65	_	33.65	_	0.81	1.11	2.64	−0.83	0.67	1.12	1.03	0.28
No. of spikelets (Nos.)	168.26	171.81	163.99	189.93 ^b^	165.66	176.83	163.33	190.16	170.33	_	170.3	165.66	0.29	−0.02	0.48	0.45	0.27	−0.18	0.27	0.12
Unfilled grain (Nos.)	33.95	29.2	30.08	47.65 ^b^	31	26.83	27.66	44	27.66	_	27.66	44	7.58	1.27	0.01	5.43	1.85	1.08	0.68	1.81
Filled grain (Nos.)	134.3	142.61	133.91	142.27	134.66	150	134.66	150.5	133	_	133	156	0.5	−0.24	1.02	−0.07	0.39	−0.37	0.48	−0.34
Panicle weight (g)	3.32	3.52	3.29	3.61	3.23	3.55	3.23	3.67	3.06	_	3.18	2.98	6.04	−0.03	7.09	0.27	1.37	0.29	1.52	0.1
100 seed weight (g)	2.3	2.27	2.27	2.4	2.35	2.26	2.32	2.39	2.38	2.2	2.38	_	1.28	1.22	1.33	1.41	−0.81	0.15	−0.73	−0.78

_ Indicates not determined, bold values indicate statistically significant mean difference between the different alleles of *CYP71P6*. ^a^—Significance at 5% level of significance, ^b^—significance at 1% level of significance. The Z test was used for the mean difference analysis.

**Table 5 plants-12-03035-t005:** The population and haplotype mean difference analysis for yield-related traits in *OsCYP71P6* in/dels rice variants.

Primer Name	Traits ^a^	Mean ± SD, AL320 ^b^	Mean ± SD, AL350 ^b^	*p* Value ^c^		
*OsCYP71P6-1* (In varieties)	NT	10.29 ± 3.09	11.98 ± 4.09	0.04, *		
	SPY	31.21 ± 11.42	37.90 ± 15.38	0.03, *		
*OsCYP71P6-4* (In varieties)	Traits	Mean ± SD, AL380 ^b^	Mean ± SD, AL400 ^b^	*p* value		
	SPY	30.71 ± 11.38	38.48 ± 13.87	0.005, **		
	NS	163.99 ± 54.16	189.93 ± 43.86	0.004, **		
	PW	3.29 ± 0.95	3.61 ± 0.84	0.05, *		
	PL	25.89 ± 2.81	27.36 ± 2.26	0.002, **		
	UG	30.08 ± 15.86	47.65 ± 28.66	0.001, **		
*OsCYP71P6-1* and *OsCYP71P6-4* (In two subpopulations)	Traits	Mean ± SD, Sub-Pop1	Mean ± SD, Sub-Pop2	*p* value		
	SPY	37.06 ± 15.33	31.42 ± 10.48	0.03, *		
	FG	150.93 ± 28.71	135.32 ± 43.63	0.02, *		
*OsCYP71P6-1* and *OsCYP71P6-4* (In two subpopulations and admixtures)	Trait	Mean ± SD, Sub-Pop1	Mean ± SD, Sub-Pop1	Mean ± SD, Admix	*p* value	
	SPY	37.06 ± 15.33	31.42 ± 10.48	29.74 ± 11.37	0.03, *	
	NS	180.12 ± 35.8	172.97 ± 49.09	154.85 ± 42.23	0.04, *	
	FG	150.93 ± 28.71	135.32 ± 43.63	124.92 ± 35.77	0.02, *	
*OsCYP71P6-1* and *OsCYP71P6-4* (In four haplotypes)	Traits	Mean ± SD, Hap1	Mean ± SD, Hap2	Mean ± SD, Hap3	Mean ± SD, Hap4	*p* value
	SPY	29.81 ± 10.24	37.71 ± 14.41	36.64 ± 16.43	42.32 ± 11.73	0.005, **

^a^—NT-No. of tillers, SPY—single-plant yield, NS—no. of spikelets, PW—panicle weight, PL—panicle length, UG—no. of unfilled grains, FG—no. of filled grains, ^b^—AL-amplicon length of the in/del primer, ^c^—Z test *p* value for mean difference. Double asterisk indicates 1% level of significance and single asterisk indicates significance at 5% level.

**Table 6 plants-12-03035-t006:** Linear regression analysis for the association of genetic variants of *OsCYP71P6* in the 3K database with spikelet fertility score in rice.

Sl.No	SNP Position ^a^	Gene Position	Amino Acid Substitution	*p* Value ^d^
1	Chr12:9581604	First Exon (98C>T) ^b^	Ser33Leu ^c^	0.005767 **
2	Chr12: 9582455	Promoter	-	0.003087 **
3	Chr12: 9582489	Promoter	-	0.019201 *
4	Chr12: 9582557	Promoter	-	0.047372 *
5	Chr12: 9582591	Promoter	-	0.004385 **
6	Chr12: 9582869	Promoter	-	0.001087 **
7	Chr12: 9583776	Promoter	-	0.020460 *
8	Chr12: 9583083	Promoter	-	0.027545 *
9	Chr12: 9582921	Promoter	-	0.003404 **

^a^—SNP position indicates the nucleotide position significantly associated with the spikelet fertility score, ^b^—98C>T indicates the 98th nucleotide substitution in the coding sequence from the start codon of CYP71A1, ^c^—serine is substituted by leucine amino acid in the 33rd position of the protein, ^d^—double asterisk indicates significance at 1% level of significance and single asterisk indicates significance at 5% level.

## Data Availability

Data is available in the manuscript and in the Appendix A.

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
