# Peer review of "A Comprehensive Genome-Wide Investigation of the Cytochrome 71 (OsCYP71) Gene Family: Revealing the Impact of Promoter and Gene Variants (Ser33Leu) of OsCYP71P6 on Yield-Related Traits in Indica Rice (Oryza sativa L.)"

_plants, 2023, doi:10.3390/plants12173035_

Round 1

Reviewer 1 Report

The manuscript by Sahoo et al. provided a comprehensive profile of cytochrome 71 gene family in rice and demonstrated that the promoter variants of one gene in this family correlated with yield traits.  Overall the paper is well written and informative.

Comments:

1. Did the author detected haplotype variation in rice population for other cytochrome 71 family genes? 

2. Does the gene expression of OsCYP71P6 changed for different haplotypes?   

3. Please included more information for the OsCYP71P6 gene from literatures especially on the yield related side. 

4. line 186, missing value of Ka/Ks score used for indicating purifying selection.

5. Could the authors provide rice MSU gene ID (e.g. Loc_Os##g####) or IRGSP ID (e.g OsXXg######) to for the identified cytochrome 71 family genes for attracting broader attendance from audience?

6.Figure 10, please specify haplotype for 'Hap1', 'Hap2', 'Hap3', and 'Hap4' in either figure legend or replace the label in figure. Same as it is section 2.10, please specify 'Hap2', 'Hap4', etc. as elsewhere in the paper used allele InDel  information to indicate each haplotype. 

7. Figure 7 missing 'A', 'B', 'C' labels.

Author Response

Reviewer 1

The manuscript by Sahoo et al. provided a comprehensive profile of cytochrome 71 gene family in rice and demonstrated that the promoter variants of one gene in this family correlated with yield traits.  Overall the paper is well written and informative.

Response: We sincerely appreciate your kind and thoughtful feedback on our manuscript. Your positive and constructive comments have been of great value to us, aiding us in the enhancement, refinement and improve the quality of our paper. We have meticulously reviewed your comments and have incorporated the necessary changes, marked in red using tracked changes. We trust that these revisions align with your expectations. Additionally, we have provided point-by-point responses to your comments below.

Minor comments

Comment 1: Did the author detected haplotype variation in rice population for other cytochrome 71 family genes? 

Response: Thank you for your valuable suggestion. Your input is greatly appreciated. Haplotype variation was solely identified within the OsCYP71P6 gene, not in other members of the OsCYP71 gene family.

Comment 2: Does the gene expression of OsCYP71P6 changed for different haplotypes? Response: We greatly appreciate your valuable suggestion. The current study did not delve into analyzing the expression differences among the haplotypes. Our forthcoming research endeavors will be dedicated to investigating this aspect.

Comment 3: Please included more information for the OsCYP71P6 gene from literatures especially on the yield related side. 

Response: Thank you for your valuable suggestions. In accordance with your advice, we have included additional information about the OsCYP71P6 gene's impact on yield-related traits. These additions have been highlighted in red within the manuscript.

Comment 4: line 186, missing value of Ka/Ks score used for indicating purifying selection.

Response: Thanks, as per reviewer's suggestions, we corrected it and mark in red color in the manuscript.

Comment 5: Could the authors provide rice MSU gene ID (e.g. Loc_Os##g####) or IRGSP ID (e.g OsXXg######) to for the identified cytochrome 71 family genes for attracting broader attendance from audience?

Response: I appreciate your valuable suggestion. In this study, we successfully identified a total of 105 OsCYP71 family genes within the genome of indica rice. Notably, the gene IDs commence with "BGIOSGA001610." It's worth mentioning that the gene IDs that begin with "LOC_Os" and "OsXXG" are characteristic of Japonica rice.

Comment 6: Figure 10, please specify haplotype for 'Hap1', 'Hap2', 'Hap3', and 'Hap4' in either figure legend or replace the label in figure. Same as it is section 2.10, please specify 'Hap2', 'Hap4', etc. as elsewhere in the paper used allele InDel information to indicate each haplotype. 

Response: We appreciate your valuable input. The amplicon sizes of the four haplotypes in rice varieties have been included in the Figure legends, following your suggestion. Similarly, the sequence details of the haplotypes are provided in Table S10. These six haplotypes were initially identified in the CYP71P6 gene through the SNP seek database, and their sequence information is accessible in Table S10. Notably, the modifications have been highlighted in the revised manuscript using blue text color.

Comment 7: Figure 7 missing 'A', 'B', 'C' labels.

Response: Thanks, We have incorporated the A, B, and C labels into Figure 7 and indicated the changes in red within the manuscript.

Reviewer 2 Report

The supplementary files and no published material in the submission system were able to download, but it showed that the file is corrupted and cannot be opened. The following issues were found only basing on V1 version.

1.    The number of genes shown on the chromosomes in figure 2 does not match the corresponding text, for example, in line 169-170, “the chromosome 7 had the most OsCYP71 genes (20 genes) and the chromosome 6 had the least OsCYP71 genes (1 gene) ” should change into “the chromosome 2 had the most OsCYP71 genes (20 genes) and the chromosome 7 had the least OsCYP71 genes (1 gene) ”.

2. Image is not clear such as the p-value and motif locations of figure 5A, the gene name in figure 7, 8. Please improve the clarity of the above images.

3. In table 2, the Marker column is CYP71A1-1 and CYP71A1-4, but the text is OsCYP71P6-1 and OsCYP71P6-4 in line 313314, please correct it.

4. Please explain means of the negative sign (-) in table 4. For example, the number of tillers or single plant yield is -1 in OsCYP71P6-4 line of Kurtosis. Furthermore, in OsCYP71P6-1 line of Skewness, the number of tillers and panicle length are positive, while the number of spikelets and filled grains are 0, and the 100 grains weight is positive, please explain. 

5.  The language needs to be revised or streamlined by professionals. For example, in line 276277279280281282,“in the leaf blade vegetative, leaf blade reproductive, leaf blade ripeningis recommended to change to in the vegetative, reproductive or ripening leaf bladeand so on. In 307, “24rice” change into “24 rice”; in 239-241, “Among them, we found that ARE element was found in the promoter regions of 40 genes, and the LTR element was found in the promoter regions of 38 genes with the number of 1-2 elements and TC-rich repeats elements were found in 21 gene promoter regions with 1-2 elements.” change into “Among them, ARE element was found in the promoter regions of 40 genes, the LTR element was found in the promoter regions of 38 genes with the number of 1-2 elements and TC-rich repeats elements were found in 21 gene promoter regions with 1-2 elements.” and so on.

Extensive editing of English language required

Author Response

Reviewer 2

The supplementary files and no published material in the submission system were able to download, but it showed that the file is corrupted and cannot be opened. The following issues were found only basing on V1 version.

Response: We sincerely appreciate your kind and thoughtful feedback on our manuscript. Your positive and constructive comments have been of great value to us, aiding us in the enhancement, refinement and improve the quality of our paper. We have meticulously reviewed your comments and have incorporated the necessary changes, marked in red using tracked changes. We trust that these revisions align with your expectations. Additionally, we have provided point-by-point responses to your comments below.

Minor Comments:

      Comment 1: The supplementary files and no published material in the submission system were able to download, but it showed that the file is corrupted and cannot be opened.

      Response: We sincerely apologize for any inconvenience you may have experienced, while attempting to download the supplementary files due to issues with the submission system. This time, we will upload the new files, and we are optimistic that you will be able to successfully download them.

     Comment 2: The number of genes shown on the chromosomes in figure 2 does not match the corresponding text, for example, in line 169-170, “the chromosome 7 had the most OsCYP71 genes (20 genes) and the chromosome 6 had the least OsCYP71 genes (1 gene) ” should change into “the chromosome 2 had the most OsCYP71 genes (20 genes) and the chromosome 7 had the least OsCYP71 genes (1 gene) ”.

Response: Thanks, as per reviewer's suggestions, we corrected it and mark in red color in the manuscript.

    Comment 3: Image is not clear such as the p-value and motif locations of figure 5A, the gene name in figure 7, 8. Please improve the clarity of the above images.

    Response: We have addressed the issue as per the reviewer's suggestions and highlighted the changes in red within the manuscript.

    Comment 4: In table 2, the Marker column is CYP71A1-1 and CYP71A1-4, but the text is OsCYP71P6-1 and OsCYP71P6-4 in line 313、314, please correct it.

    Response: Thanks, as per reviewer's suggestions, we corrected it and mark in red color in the manuscript.

    Comment 5: Please explain means of the negative sign (-) in table 4. For example, the number of tillers or single plant yield is -1 in OsCYP71P6-4 line of Kurtosis. Furthermore, in OsCYP71P6-1 line of Skewness, the number of tillers and panicle length are positive, while the number of spikelets and filled grains are 0, and the 100 grains weight is positive, please explain. 

    Response: Thanks for your valuable suggestion. The values has been rechecked and corrected in the Table 4. Skewness is the measure of the asymmetry and values between -0.5 to +0.5 indicates normal distribution and values < -1.0 and > +1.0 indicates negative and positive skewness. For example, skewness of 1.2 for no. of tillers indicates slightly positively skewed. Similarly, kurtosis values range between -3 to + 3 indicates normal distribution. For example, most of the traits in Table 4 show normal distribution except for unfilled grains and panicle weight. As per reviewer's suggestions, we corrected it and mark in red color in the manuscript..

    Comment 6: The language needs to be revised or streamlined by professionals. For example, in line 276、277、279、280、281、282,“in the leaf blade vegetative, leaf blade reproductive, leaf blade ripening” is recommended to change to “in the vegetative, reproductive or ripening leaf blade” and so on. In 307, “24rice” change into “24 rice”; in 239-241, “Among them, we found that ARE element was found in the promoter regions of 40 genes, and the LTR element was found in the promoter regions of 38 genes with the number of 1-2 elements and TC-rich repeats elements were found in 21 gene promoter regions with 1-2 elements.” change into “Among them, ARE element was found in the promoter regions of 40 genes, the LTR element was found in the promoter regions of 38 genes with the number of 1-2 elements and TC-rich repeats elements were found in 21 gene promoter regions with 1-2 elements.” and so on.

Response: Thanks for your valuable suggestion. As per reviewer's suggestions, we corrected it and mark in red color in the manuscript.

Reviewer 3 Report

In the submitted manuscript, Sahoo et al. comprehensively analyzed the characteristics of the Cytochrome 71 (OsCYP71) family. They found promoter allelic variation and Ser33Leu amino acid substitution of the OsCYP71P6 gene could affect the yield, providing helpful information for rice breeding. Here are my detailed comments.

1. The authors should change the title to “......Revealed that Sequence Variants of OsCYP71P6.......” since the Ser33Leu amino acid substitution is also an essential variant for the function of CYP71P6.

2. In Table 1, the authors should indicate the database of Gene ID. And what is the specific meaning of GRAVY?

3. Figure 1, what is the specific meaning of (29), (8)......? Are these numbers indicate the references? I did not catch these meanings.

4. Please provide high-resolution figures for Figure 4 (especially the X-axis), 5 (especially 5A p-value, the left panel of 5B), 7 (especially gene-ID) and 9  (especially 9A-B).

5. Table 4 and Figure 10 should add statistical significance analysis. The authors should also indicate the method used for statistical analysis.

Author Response

Reviewer 3

In the submitted manuscript, Sahoo et al. comprehensively analyzed the characteristics of the Cytochrome 71 (OsCYP71) family. They found promoter allelic variation and Ser33Leu amino acid substitution of the OsCYP71P6 gene could affect the yield, providing helpful information for rice breeding. Here are my detailed comments.

Response: We sincerely appreciate your kind and thoughtful feedback on our manuscript. Your positive and constructive comments have been of great value to us, aiding us in the enhancement, refinement and improve the quality of our paper. We have meticulously reviewed your comments and have incorporated the necessary changes, marked in red using tracked changes. We trust that these revisions align with your expectations. Additionally, we have provided point-by-point responses to your comments below.

Minor Comments:

Comment 1: The authors should change the title to “......Revealed that Sequence Variants of OsCYP71P6.......” since the Ser33Leu amino acid substitution is also an essential variant for the function of CYP71P6.

Response: Thanks for your valuable suggestion. We changed the manuscript title and marked in red color in the manuscript.

Comment 2: In Table 1, the authors should indicate the database of Gene ID. And what is the specific meaning of GRAVY?

Response: I appreciate your insightful recommendation. The material and methods section includes details about the Gene ID database. In the context of the Grand Average of Hydropathy, the acronym GRAVY is elucidated. In table 1, we explicitly provide the complete expansion of GRAVY. The calculation of the GRAVY score involves summing up the hydropathy values of all amino acids within a protein and subsequently dividing this sum by the total residue count. A negative GRAVY value signifies a non-polar protein nature, whereas a positive value indicates a polar protein characteristic. These distinctions are visually highlighted in red within the manuscript.

Comment 3: Figure 1, what is the specific meaning of (29), (8)......? Are these numbers indicate the references? I did not catch these meanings.

Response: I appreciate your valuable input. The phylogenetic tree was constructed utilizing 42 proteins sourced from A. thaliana, 105 proteins from O. sativa, and 43 proteins from S. lycopersicum. This information is not cited as a reference; you can refer to Table S2 for further clarification. We have rectified this aspect and highlighted the necessary correction in red within the manuscript.

Comment 4: Please provide high-resolution figures for Figure 4 (especially the X-axis), 5 (especially 5A p-value, the left panel of 5B), 7 (especially gene-ID) and 9  (especially 9A-B).

Response: I appreciate your insightful suggestion. We have incorporated the high-resolution figures, namely 4, 5A, 5B, 7A, 7B, 7C, 9A, and 9B and mark in red color in the manuscript.

Comment 5: Table 4 and Figure 10 should add statistical significance analysis. The authors should also indicate the method used for statistical analysis.

Response: Thanks for your valuable suggestion. We reanalyzed the data and now included the information regarding statistical significance in both Table 4 and Figure 10 and mark in red color in the manuscript. The statistical analysis was performed using the Z test for mean difference at 5% and 1% level of significance and mark in red color in the manuscript.

Round 2

Reviewer 2 Report

1.     In line 230, the legend is incomplete and it is recommended to delete it.

2.     The quality of Figure 8 has not yet improved.

Minor editing of English language

Author Response

Reviewer 2

In the submitted manuscript, Sahoo et al. comprehensively analyzed the characteristics of the Cytochrome 71 (OsCYP71) family. They found promoter allelic variation and Ser33Leu amino acid substitution of the OsCYP71P6 gene could affect the yield, providing helpful information for rice breeding. Here are my detailed comments.

Response: We sincerely appreciate your kind and thoughtful feedback on our manuscript. Your positive and constructive comments have been of great value to us, aiding us in the enhancement, refinement and improve the quality of our paper. We have meticulously reviewed your comments and have incorporated the necessary changes, marked in red using tracked changes. We trust that these revisions align with your expectations. Additionally, we have provided point-by-point responses to your comments below.

Minor Comments:

Comment 1: In line 230, the legend is incomplete and it is recommended to delete it.

Response: We appreciate your valuable suggestion. The figure legend in the manuscript title has been modified, and it has been highlighted in red within the manuscript.

Comment 2: The quality of Figure 8 has not yet improved.?

Response: I'm grateful for your perceptive suggestion. We've integrated the high-quality figure into the manuscript.